

# Reinterpretation of LHC results for new physics: status and recommendations after run 2

### The LHC BSM Reinterpretation Forum

## Abstract

We report on the status of efforts to improve the reinterpretation of searches and measurements at the LHC in terms of models for new physics, in the context of the LHC Reinterpretation Forum. We detail current experimental offerings in direct searches for new particles, measurements, technical implementations and Open Data, and provide a set of recommendations for further improving the presentation of LHC results in order to better enable reinterpretation in the future. We also provide a brief description of existing software reinterpretation frameworks and recent global analyses of new physics that make use of the current data.

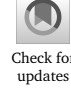
Waleed Abdallah[1,2], Shehu AbdusSalam[3], Azar Ahmadov[4], Amine Ahriche[5,6], Gaël Alguero[7], Benjamin C. Allanach[8*], Jack Y. Araz[9], Alexandre Arbey[10,11], Chiara Arina[12], Peter Athron[13], Emanuele Bagnaschi[14], Yang Bai[15], Michael J. Baker[16], Csaba Balazs[13], Daniele Barducci[17,18], Philip Bechtle[19*], Aoife Bharucha[20], Andy Buckley[21†], Jonathan Butterworth[22*], Haiying Cai[23], Claudio Campagnari[24], Cari Cesarotti[25], Marcin Chrzaszcz[26], Andrea Coccaro[27], Eric Conte[28,29], Jonathan M. Cornell[30], Louie D. Corpe[22], Matthias Danninger[31], Luc Darmé[32], Aldo Deandrea[10], Nishita Desai[33*], Barry Dillon[34], Caterina Doglioni[35], Matthew J. Dolan[16], Juhi Dutta[1,36], John R. Ellis[37], Sebastian Ellis[38], Matthew Feickert[40], Nicolas Fernandez[40], Sylvain Fichet[41], Thomas Flacke[42], Benjamin Fuks[43,44*], Achim Geiser[45], Marie-Hélène Genest[7], Akshay Ghalsasi[46], Tomas Gonzalo[13], Mark Goodsell[43], Stefania Gori[46], Philippe Gras[47], Admir Greljo[11], Diego Guadagnoli[48], Sven Heinemeyer[49,50,51], Lukas A. Heinrich[11*], Jan Heisig[12*], Deog Ki Hong[52], Tetiana Hryn'ova[53], Katri Huitu[54], Philip Ilten[55], Ahmed Ismail[56], Adil Jueid[57], Felix Kahlhoefer[58*], Jan Kalinowski[59], Jernej F. Kamenik[34,60], Deepak Kar[61], Yevgeny Kats[62], Charanjit K. Khosa[63], Valeri Khoze[64], Tobias Klingl[19], Pyungwon Ko[65], Kyoungchul Kong[66], Wojciech Kotlarski[67], Michael Krämer[58], Sabine Kraml[7¶], Suchita Kulkarni[68*], Anders Kvellestad[69,70*], Clemens Lange[11*], Kati Lassila-Perini[71], Seung J. Lee[72], Andre Lessa[73*], Zhen Liu[74], Lara Lloret Iglesias[49], Jeanette M. Lorenz[75], Danika MacDonell[76], Farvah Mahmoudi[10,11*], Judita Mamuzic[77], Andrea C. Marini[78], Pete Markowitz[79], Pablo Martinez Ruiz del Arbol[49], David Miller[80], Vasiliki A. Mitsou[77], Stefano Moretti[81,82], Marco Nardecchia[17], Siavash Neshatpour[10], Dao Thi Nhung[83], Per Osland[84], Patrick H. Owen[85*], Orlando Panella[86], Alexander Pankov[87], Myeonghun Park[88], Werner Porod[89], Darren D. Price[90*], Harrison Prosper[91], Are Raklev[69*], Jürgen Reuter[45], Humberto Reyes-González[7], Thomas Rizzo[38], Tania Robens[92], Juan Rojo[93], Janusz Andrzej Rosiek[59], Oleg Ruchayskiy[94], Veronica Sanz[63,77], Kai Schmidt-Hoberg[45], Pat Scott[70,95‡], Sezen Sekmen[96*], Dipan Sengupta[97], Elizabeth Sexton-Kennedy[98],

Hua-Sheng Shao[43], Seodong Shin[99], Luca Silvestrini[11,18], Ritesh Singh[100], Sukanya Sinha[61], Jory Sonneveld[101], Tim Stefaniak[45], Yotam Soreq[102], Giordon H. Stark[46], Jesse Thaler[103], Riccardo Torre[11,27], Emilio Torrente-Lujan[104], Gokhan Unel[105], Natascia Vignaroli[106], Wolfgang Waltenberger[68⋆], Nicholas Wardle[70°], Graeme Watt[64], Georg Weiglein[45], Martin J. White[107], Sophie L. Williamson[108], Jonas Wittbrodt[109], Lei Wu[110], Stefan Wunsch[11], Tevong You[8,11,111], Yang Zhang[13] and José Zurita[112,113]

**1** Harish-Chandra Research Institute (HBNI), Allahabad 211019, India
**2** Department of Mathematics, Faculty of Science, Cairo University, Giza 12613, Egypt
**3** Department of Physics, Shahid Beheshti University, Tehran, Islamic Republic of Iran
**4** Department of Theoretical Physics, Baku State University, AZ-1148 Baku, Azerbaijan
**5** The Abdus Salam International Centre for Theoretical Physics, I-34014, Trieste, Italy
**6** Laboratoire de Physique des Particules et Physique Statistique,
ENS, DZ-16050 Algiers, Algeria
**7** Univ. Grenoble Alpes, CNRS, Grenoble INP, LPSC-IN2P3, 38000 Grenoble, France
**8** DAMTP, University of Cambridge, Cambridge, CB3 0WA, UK
**9** Concordia University, 7141 Sherbrooke St. West, Montréal, QC, Canada H4B 1R6
**10** Université de Lyon 1, CNRS/IN2P3, UMR 5822 IP2I, 69622 Villeurbanne, France
**11** CERN, European Organization for Nuclear Research, Geneva, Switzerland
**12** CP3, Université catholique de Louvain, B-1348 Louvain-la-Neuve, Belgium
**13** School of Physics and Astronomy, Monash University, Melbourne, Victoria 3800, Australia
**14** Paul Scherrer Institut, CH-5232 Villigen PSI, Switzerland
**15** Department of Physics, University of Wisconsin-Madison, Madison, WI 53706, USA
**16** School of Physics, The University of Melbourne, Victoria 3010, Australia
**17** Sapienza - Università di Roma, Piazzale Aldo Moro 2, 00185, Roma, Italy
**18** INFN, Sezione di Roma, Piazzale Aldo Moro 2, 00185, Roma, Italy
**19** Universität Bonn, Physikalisches Institut, Nussallee 12, 53115 Bonn, Germany
**20** CPT, UMR7332, CNRS and Aix-Marseille Université and Université de Toulon
13288 Marseille, France
**21** School of Physics & Astronomy, University of Glasgow, Glasgow, G12 8QQ, UK
**22** Department of Physics & Astronomy, University College London, London, WC1E 6BT, UK
**23** Asia Pacific Center for Theoretical Physics, Pohang, Gyeongbuk 790-784, South Korea
**24** Department of Physics, University of California, Santa Barbara, CA 93016, USA
**25** Harvard University, Center for the Fundamental Laws of Nature,
Cambridge MA, 02143, USA
**26** Henryk Niewodniczanski Institute of Nuclear Physics
Polish Academy of Sciences, Krakow, Poland
**27** INFN, Sezione di Genova, Via Dodecaneso 33, I-16146 Genova, Italy
**28** Université de Strasbourg, CNRS, IPHC UMR 7178, Strasbourg, France
**29** Université de Haute-Alsace, Mulhouse, France
**30** Department of Physics, University of Cincinnati, Cincinnati, Ohio 45221, USA
**31** Department of Physics, Simon Fraser University, Burnaby, BC, Canada V5A 1S6
**32** INFN, Laboratori Nazionali di Frascati, C.P. 13, 100044 Frascati, Italy
**33** Tata Institute of Fundamental Research, Mumbai 400005, India
**34** Jožef Stefan Institute, Jamova Cesta 39, 1000 Ljubljana, Slovenia
**35** Fysikum, Lund University, Professorsgatan 1, 22263 Lund, Sweden
**36** II. Institut für Theoretische Physik, Universität Hamburg, 22761 Hamburg, Germany
**37** Department of Physics, King's College London, London WC2R 2LS, UK
**38** SLAC National Accelerator Laboratory, Stanford University, Menlo Park, CA, USA
**39** Mohammed V University of Rabat, B.P.8007.N.U, Agdal, Morocco
**40** Department of Physics, University of Illinois at Urbana-Champaign,

Urbana, IL 61801, USA

41 ICTP South American Institute for Fundamental Research & IFT-UNESP, São Paulo, Brazil

42 Center for Theoretical Physics of the Universe, IBS, Daejeon 34126, South Korea

43 LPTHE, UMR 7589, Sorbonne Université et CNRS, 75252 Paris Cedex 05, France

44 Institut Universitaire de France, 75005 Paris, France

45 Deutsches Elektronen-Synchrotron DESY, Notkestr. 85, 22607 Hamburg, Germany

46 Santa Cruz Institute for Particle Physics, UC Santa Cruz, CA 95064, USA

47 IRFU, CEA, Université Paris-Saclay, Gif-sur-Yvette, France

48 LAPTh, CNRS, USMB et UGA, 74941 Annecy-le-Vieux Cedex, France

49 Instituto de Física de Cantabria (CSIC-UC), 39005 Santander, Spain

50 Spain Campus of International Excellence UAM+CSIC, Cantoblanco, 28049 Madrid, Spain

51 Instituto de Física Teórica UAM-CSIC, 28049 Madrid, Spain

52 Department of Physics, Pusan National University, Busan 46241, South Korea

53 Univ. Grenoble Alpes, Univ. Savoie Mont Blanc, CNRS, IN2P3-LAPP, Annecy, France

54 Department of Physics and Helsinki Institute of Physics,
00014 University of Helsinki, Finland

55 School of Physics and Astronomy, University of Birmingham, Birmingham, UK

56 Department of Physics, Oklahoma State University, Stillwater, OK 74078, USA

57 Department of Physics, Konkuk University, Seoul 05029, Republic of Korea

58 TTK, RWTH Aachen University, D-52056 Aachen, Germany

59 Faculty of Physics, University of Warsaw, 02-093 Warsaw, Poland

60 Faculty of Mathematics and Physics, University of Ljubljana, 1000 Ljubljana, Slovenia

61 School of Physics, University of Witwatersrand, Johannesburg, South Africa

62 Department of Physics, Ben-Gurion University, Beer-Sheva 8410501, Israel

63 Department of Physics and Astronomy, University of Sussex, Brighton BN1 9QH, UK

64 IPPP, Department of Physics, Durham University, Durham, DH1 3LE, UK

65 School of Physics, Korea Institute for Advanced Study, Seoul 02455, South Korea

66 Department of Physics and Astronomy, University of Kansas, Lawrence, KS 66045, USA

67 Institut für Kern- und Teilchenphysik, TU Dresden, 01069 Dresden, Germany

68 Institut für Hochenergiephysik, Österreichische Akademie der Wissenschaften,
1050 Wien, Austria

69 Department of Physics, University of Oslo, N-0316 Oslo, Norway

70 Department of Physics, Imperial College London, London SW7 2AZ, UK

71 Helsinki Institute of Physics, Finland

72 Department of Physics, Korea University, Seoul 136-713, South Korea

73 Centro de Ciências Naturais e Humanas, UFABC Santo André, 09210-580 SP, Brazil

74 Maryland Center for Fundamental Physics, Univ. of Maryland,
College Park, MD 20742, USA

75 Ludwig Maximilians Universität, Am Coulombwall 1, 85748 Garching, Germany

76 University of Victoria, Victoria, V8P 5C2, Canada

77 Instituto de Física Corpuscular, CSIC, Universitat de València, 46980 Paterna, Spain

78 Department of Physics, Massachusetts Institute of Technology,
Cambridge, MA 02139, USA

79 Florida International University, Miami FL 33199, USA

80 The Enrico Fermi Institute and The University of Chicago, Chicago, IL 60637, USA

81 School of Physics & Astronomy, University of Southampton,
Highfield, Southampton SO17 1BJ, UK

82 Particle Physics Department, STFC Rutherford Appleton Laboratory, Oxon OX11 0QX, UK

83 Institute for Interdisciplinary Research in Science and Education,
ICISE, 590000 Quy Nhon, Vietnam



84 Department of Physics and Technology, University of Bergen, N-5020 Bergen, Norway
85 Physik-Institut, Universität Zürich, CH-8057, Switzerland
86 INFN, Sezione di Perugia, Perugia, I-06123, Italy
87 Joint Institute for Nuclear Research, Dubna, Russia
88 Institute of Convergence Fundamental Studies & School of Liberal Arts,
Seoultech, Seoul 01811, Korea
89 Institut für Theoretische Physik und Astrophysik, Universität Würzburg, Germany
90 Department of Physics and Astronomy, University of Manchester,
Manchester, M13 9PL, UK
91 Department of Physics, Florida State University, FL 32306, USA
92 Ruder Boskovic Institute, Bijenicka cesta 54, 10000 Zagreb, Croatia
93 Department of Physics and Astronomy, VU Amsterdam, 1081 HV Amsterdam, NL
94 Niels Bohr Institute, Copenhagen University, Copenhagen, DK 2100, Denmark
95 School of Mathematics and Physics, University of Queensland,
Brisbane QLD 4072, Australia
96 Department of Physics, Kyungpook National University, Daegu, South Korea
97 Department of Physics and Astronomy, University of California San Diego, San Diego, USA
98 Fermi National Accelerator Laboratory, Batavia, IL 60510, USA
99 Department of Physics, Jeonbuk National University, Jeonju, Jeonbuk 54896, South Korea
100 Department of Physical Sciences, IISER Kolkata, Mohanpur, 741246, India
101 Institut für Experimentalphysik, Universität Hamburg, 22761 Hamburg, Germany
102 Physics Department, Technion—Israel Institute of Technology, Haifa 3200003, Israel
103 Center for Theoretical Physics, Massachusetts Institute of Technology,
Cambridge, MA 02139, USA
104 IFT, Dept. Physics, Universidad de Murcia, 30100 Murcia, Spain
105 University of California at Irvine, Department of Physics and Astronomy, Irvine, USA
106 Dipartimento di Fisica "E. Fermi", Universitá di Pisa and INFN Pisa, 56127, Pisa, Italy
107 University of Adelaide, North Terrace Adelaide SA 5034, Australia
108 Institute for Theoretical Physics, Karlsruhe Institute of Technology,
76128 Karlsruhe, Germany
109 Department of Astronomy and Theoretical Physics,
Lund University, 22362 Lund, Sweden
110 Dep. of Physics & Inst. of Theoretical Physics,
Nanjing Normal University, Nanjing 210023, China
111 Cavendish Laboratory, University of Cambridge, Cambridge, CB3 0HE, UK
112 Inst. for Nuclear Physics, Karlsruhe Institute of Technology,
76344 Eggenstein-Leopoldshafen, Germany
113 Institute for Theoretical Particle Physics, Karlsruhe Institute of Technology,
76128 Karlsruhe, Germany

⋆ Editor
† Lead Editor: Andy.Buckley@glasgow.ac.uk
¶ Lead Editor: sabine.kraml@lpsc.in2p3.fr
‡ Lead Editor: pat.scott@uq.edu.au
∘ Lead Editor: n.wardle09@imperial.ac.uk

# Contents

# 1 Introduction

The LHC experiments ATLAS, CMS, and LHCb each perform precise measurements of Standard Model (SM) processes as well as direct searches for physics beyond the SM (BSM) in a vast variety of channels. Despite the multitude of BSM scenarios tested this way by the experiments, this still constitutes only a small subset of the possible theories and parameter combinations to which the experiments are sensitive. In many cases, the subjects of official interpretations by the experimental collaborations are so-called simplified models, designed to facilitate searches for a wide range of possible signatures of new physics, but ultimately unable to capture the

full phenomenology of the ultra-violet complete models from which they may derive.

In order to determine the implications of LHC data for a broad range of theories, experimental analyses should be reinterpretable in terms of theories not considered in the original analysis publication. This reinterpretation process, also known as "recasting", makes it possible for the community as a whole to test a much larger variety of theories using the LHC than would be possible purely within the experimental collaborations. This also makes it possible for phenomenologists to give detailed feedback on the original analyses, and to better suggest promising new avenues of experimental analysis. Reinterpretation is only possible with the provision of detailed analysis information by the experiments: the more detailed this information, the broader and more accurate the types of reinterpretation that are enabled. A number of major public reinterpretation software packages have been developed to make use of this information.

While acknowledging that ensuring reinterpretability requires significant time involvement, the experimental collaborations also benefit from this investment. Results of reinterpretation studies aid the experiments in identifying target BSM scenarios for future searches. Reinterpretability of a data analysis preserves its shelf life, and increases its scientific impact through derived works. Preparation of the public data products required for reinterpretation also inevitably makes *internal* reproduction easier for experimental collaborations (e.g., after a student responsible for an analysis has moved to a different position or field). Finally, provision of such data further helps to meet increasingly stringent funders' requirements for openness and reproducibility of publicly-funded research.

Following previous initiatives, notably within the Les Houches "Physics at TeV Colliders" workshop series, which resulted in a first set of recommendations in 2012 [1, 2], our community established a dedicated *"Forum on the (re-)interpretation of LHC results for BSM studies"* in 2016 [1] (hereafter referred to as the LHC Re-Interpretation Forum), with the purpose of deepening ongoing dialogue and collaboration between experimentalists and theorists working to facilitate and perform BSM reinterpretation studies. This document serves as a status report on that goal, and puts forth a set of further recommendations for improving the presentation of LHC results for reinterpretation.[2]

The current report appears after the first five LHC Re-Interpretation Forum workshops at CERN, Fermilab and Imperial College London between 2016 and 2019, at the point where the Run 2 legacy results are being prepared. We begin in Section 2 by detailing the specific data products currently available from the experiments for reinterpretation, and giving recommendations for their further improvement. This section covers direct BSM searches, measurements, technical implementations and Open Data. We then discuss in Section 3 different reinterpretation methods and the public software frameworks that enable them. Section 4 gives some examples of global phenomenological analyses of BSM models enabled by the data currently available for reinterpretation. We conclude in Section 5 with a short summary of the Forum's current recommendations.

## 2 Information provided by experiments

As the LHC programme has matured, the volume and variety of information provided for public re-use by the experimental collaborations have increased markedly, reflecting dialogues with the SM and BSM phenomenology communities. For usability, the availability of this information in electronic format is crucial. The main distribution mechanisms for both measurement

---

[1]https://twiki.cern.ch/twiki/bin/view/LHCPhysics/InterpretingLHCresults

[2]In a similar spirit, recommendations for the presentation of results were elaborated by the LHC Dark Matter Forum in [3] and the LHC Long-Lived Particles Community in Ref. [4].

and search data are the HEPData [5] database and the experiments' own web pages. Moreover, important initiatives are ongoing to provide open access to the full data set (after some time-lag) [6,7] and to respond in a semi-automated way to requests for re-analysis [8], to which we return in later sections.

The desire of the experimental collaborations to be conservative about making analysis data public until the analysis is fully confirmed, by either paper submission or acceptance, is understandable. However, experience has shown that if figures are presented in public and if there is a long timescale between a preliminary and a published result, they will be (imperfectly) digitised for use in reinterpretations. Moreover, journals have shown no reluctance to publish phenomenological studies that rely on data labelled "preliminary". We suggest that it is better to make numerical data available and for external users to be clear in turn that results based on preliminary data remain preliminary themselves, until all inputs are confirmed. This could also help the collaborations get feedback from the community on improvements, which could be made between the preliminary and published results.

An "early and often" strategy, combined with official encouragement and even formal requirement for publication, makes it more likely that numerical data *will* be published: at present, the process too often fails to complete in general purpose experiments. We note however that the LHCb collaboration usually provides numerical results via collaboration public wiki pages within a few days of the release of an analysis.

## 2.1 Direct BSM searches

The information provided by the search analyses typically falls into the following categories:

- Analysis description: detailed description of the analysis strategy, including the definition of objects and kinematic variables used, signal selection criteria, etc. This is nowadays done in a clear and explicit way in most publications. Open questions however remain e.g. for analyses which use non-standard objects or employ machine learning (ML) techniques.

- Primary data: typically signal-region event counts and/or kinematic distributions. These are available in papers and available electronically with an increasing frequency. With substantial effort, they may also be (re)derivable from Open Data, when available (Section 2.3).

- Background estimates: provided in papers and frequently, but not universally, published electronically for search signal regions (where they are essential to reinterpretation). Again, may be rederivable from Open Data.

- Correlations: second-order correlation data as Simplified Likelihood [9] covariance or correlation matrices are provided by CMS in some cases, typically via auxiliary data stores; ATLAS does not usually provide explicit search-region correlation data, but has recently started to publish full likelihoods from which they could be extracted (see below).

- Smearing functions and efficiencies: Generic trigger and reconstruction efficiencies and resolution performance data are found in a variety of detector performance papers, but often long after the relevant analysis papers, or in public figures available on the collaboration webpages; in both cases, they are not always in a digitised or executable form. Occasionally, analysis-specific resolution data are provided, e.g. the ATLAS 13 TeV 139 fb$^{-1}$ dilepton resonance search [10] provided parametrisations of resolution function parameters as functions of $m_{\ell\ell}$. Many of the recent searches for long-lived particles (LLPs) also provide analysis-specific efficiencies.

- Interpretation in the context of simplified models: either in the form of 95% confidence level (CL) upper limits on the signal cross-section, or as (signal) efficiency maps. In a very few cases, efficiency maps per signal regions are given in combination with correlations (see above).

- Full likelihoods: recently ATLAS has begun providing full likelihoods [11] which report $L(\theta|\mathcal{D})$, in which $\theta$ is the union of parameters of interest and possible nuisance parameters and $\mathcal{D}$ denotes the observed data. Many other types of data such as background estimates, correlations and primary data and simplified model estimates are encoded in the likelihood.

- Statistical method: information on the statistical procedure used to treat nuisance parameters and to calculate exclusion limits. For a frequentist hypothesis test this includes the definition of the test statistic and information on how the distribution of the test statistic is determined (e.g. by simulating pseudo-experiments or by using the asymptotic limit as an approximation).

- Reproduction metadata: information such as cut-flow tables and model/Monte Carlo Event Generator (MCEG) configuration files are often provided, but with large variations in content and format even within single experimental working groups. Analysis pseudocode, providing an encoding of analysis logic, has also begun to be provided by some analyses but there is not yet any general policy.

We now visit each type of (auxiliary) data in turn, discussing the use of this information and giving recommendations on how best it might be presented. We start with background estimates and come back to open questions regarding analysis description at the end of this section. For the sake of brevity, some purely technical discussions and recommendations are relegated to the appendix.

### 2.1.1 Background estimates

Background estimates are universally provided in search papers, but have only more recently begun to be reported to HEPData. They are not as timeless as the experimental data, as modelling will undoubtedly improve, but construction of the complex and CPU-expensive Standard Model MCEG samples used as inputs to most such estimates is typically beyond the means of BSM reinterpretation groups. Additionally, the final estimates are usually at least to some extent data-driven, via reweighting and profile fitting procedures unavailable outside the experiments. The experiments' best estimates of background rates at analysis time are hence crucial for reinterpretation. We also encourage decomposition of the background into separate major contributions, to enable (pre-fit) future replacement of individual process types. This would be particularly powerful if coupled with a full (perhaps simplified) likelihood publication.

Practically, as HEPData tables can have multiple "$y$-axes" (i.e. columns of measured dependent variables), we suggest that background estimates are always reported using this feature. This ensures that background estimates are unambiguously identified with their data counterparts, and that the best estimate of the total background is always available. We propose that it be identified with a standard column heading, e.g. "BKG_TOT".

A special case are so-called "bump hunts", where one searches for a sharp feature on top of a smooth background. In this case a detailed modelling of the background is not necessary and in fact often impossible. Instead, the background is fitted by a smooth function and the residuals are used to test the signal hypothesis. This approach is typically highly suitable for reinterpretation, provided that all necessary details on the fitting procedure are given. These include the definition of the fitting function, the range over which the fit is performed and the

statistical method used for the fit. It is encouraged to also publish the best-fit values of the fitted parameters together with their uncertainties; or, if this is not feasible (e.g. for sliding windows fits) the fitted value at each point with its uncertainty.

In summary, we recommend that all experimental searches provide:

1. estimates of background rates in all signal regions, broken down into as many separate process contributions as possible; and

2. where backgrounds are fitted, full details of the fits: functional forms, fit ranges, fit procedures, best fits and uncertainties.

### 2.1.2 Correlations

Correlation data, particularly the Simplified Likelihood [9] information from CMS, has proven excellent for stabilising and ensuring better statistical definition in global fits[3] as well as avoiding either overly conservative or over-enthusiastic interpretations. Currently, correlations are reported as a mixture of error sources and dedicated correlation/covariance matrix tables. We recommend standard publishing of covariance or error-source information between signal regions, in the following order of preference:

1. via a decomposition into orthogonal error sources as part of the primary dataset of signal-region yields,

2. as a separate covariance matrix, or

3. as a separate correlation matrix.

Similar considerations apply to correlations reported for measurements, cf. Section 2.2. Detailed technical considerations and recommendations on the above are given in Appendix A.

If non-Gaussian effects are relevant, we encourage the publication of correlation data using the (next-to-)simplified likelihood framework presented in Ref. [13]; this is a simple method to encode asymmetry information into correlations via publication of only $N_{\text{bins}}$ additional numbers (as opposed to the more common $N_{\text{bins}} \times N_{\text{bins}}$ second order correlation data).

### 2.1.3 Smearing functions and efficiencies

Smearing functions and efficiencies are reported in several forms. The CMS SUSY group have maintained a page linking to $(p_T, \eta)$ 2D reconstruction efficiency tabulations since the 2017 Moriond conferences [14] (these are partially specific to groups of analyses using certain reconstruction working points and kinematic phase-spaces), as well as official CMS steering files for Delphes [15]. Functional parametrisations of analysis-specific kinematic smearings have also sometimes been published, e.g. in the ATLAS 2019 139 fb$^{-1}$ dilepton resonance search analysis [10]. There is not yet a standard digitised format for tabulated or parametrised numerical efficiency and resolution data, nor do performance notes issue such data as standard.

As the ATLAS dilepton example [10] shows, functional parametrisations can be very complex, and the format in which functional forms and coefficients is provided makes a major difference to usability. Later iterations of the HEPData record of this analysis have replaced native HEPData datasets of function coefficients with source code snippets, which is more directly useful for this sort of data. Reformatting ROOT file efficiency tabulations, e.g. in

---

[3]For example, the GAMBIT EWMSSM study [12] showed that the use of single best-expected signal regions was numerically unstable as well as being statistically suboptimal. Furthermore, any approach forced to conservatively use single best-expected signal regions invalidates the interpretation of the profile log-likelihood-ratio via Wilks' Theorem, necessitating the uptake of approximate methods.

($p_T$, $|\eta|$), into a directly usable form can also require significant amounts of work, so having these data in a standard generic form (e.g. CSV, YAML or JSON) greatly assists reuse. Providing such tabulations as native HEPData datasets automatically enables such representations, and is strongly recommended. However, full standardisation of data formats needs to consider not just the container format (e.g. "ROOT", "HEPData", or "JSON") but also the structure of paths and the type of data object used within the container. Such representations are readily fit for translation either into smearing-based approaches to detector parametrisation [16, 17] or as a reference for a Delphes-based approach [18, 19].

Long-lived particle (LLP) searches are a developing area [4] in which more information is needed to capture the effects of non-standard physics-objects, such as displaced vertices or highly ionizing tracks. Indeed, for LLP searches efficiencies may be functions of several new parameters, e.g. the trigger efficiency may depend on the opening angle between two muons for a displaced dimuon analysis or the transverse displacement $L_{xy}$ of the dimuon vertex. In such cases, the respective efficiencies should be provided in the form of $n$-dimensional maps as a function of the relevant variables. Moreover, it can be very helpful to have broken-down individual efficiency information (e.g. for the trigger, identification and reconstruction) rather than a single map where all of the efficiencies are combined together.

To get around the problem of non-standard selections in LLP searches, the experimental analyses currently provide simplified signal efficiencies on truth-level MC events for their signal benchmarks. To do this, the selection efficiency must be unweighted assuming certain masses (and consequently boosts) of intermediate LLPs, and is therefore potentially model-dependent. In analyses where truth-based efficiency information is provided, it is possible to replicate the final event selection for the benchmark model quite well [20]. However, it is not completely clear whether such unweighted efficiency parametrisations can be used to accurately constrain models with event topologies that differ from that of the corresponding benchmark. To mitigate wild overestimations of reach, it was recommended in Ref. [4] that (1) efficiency be provided at "object-level" rather than event level, (2) if this is not possible, every analysis should use (at least) two benchmarks with different topologies and/or kinematics, and (3) the analysis should make public how well the simplified efficiency parametrisation reproduces the published expected signal numbers.

For a detailed discussion and recommendations specific for LLP searches, we refer the reader to Chapter 6 of [4]. To summarise the conclusion from [20, 21], per-object efficiency maps are far more accurate and general purpose than parametrisations in high-level properties of benchmark BSM models. Many recent analyses [22–24] have provided such maps in digital format, but schemes for community-standard HEPData publication of high-dimensional efficiency maps still need to be explored.

Trigger and reconstruction efficiencies are crucial information, and we urge the collaborations to make them available in digital form. This also concerns figures from performance notes, which are not published on HEPData so far.

We recommend that each experimental analysis identifies a set of efficiencies and resolution functions for its reinterpretation, and publishes them, if they are not already publicly available, in the following forms, with a quantitative measure of how well the expected signal numbers are reproduced with these:

- *Resolutions*: Delphes cards, functional parameterisations with relevant parameters (for simple functions), or code snippets (for more complex functions).

- *Efficiencies*: tabulated efficiencies in the kinematic variables most relevant for the analysis in question, in HEPData format (not ROOT format). Preferably broken down into as many sub-efficiencies (trigger, ID, etc) as possible, and given at object rather than event level.

### 2.1.4 Full likelihoods

A necessary step for reinterpretation is the construction of a statistical model, or *likelihood*, to compare the observed data to the target theory. In fact, many of the data products discussed here, such as signal/background yields and correlations, are used by the various external reinterpretation packages to construct likelihoods. Whilst extremely useful, the likelihoods constructed from these products are however always only an approximation to the true underlying experimental likelihood. The reinterpretation workflow can be greatly facilitated and rendered much more precise if the original likelihood of the analysis is published in full. We strongly encourage the movement towards the publication of full experimental likelihoods wherever possible.

LHCb has already for some time published analysis-specific full likelihood functions (e.g. [25]), many of which are now captured and publicly disseminated within the HEPLike package [26].

ATLAS has recently started to do this using a JSON serialisation of the likelihood [11], which provides background estimates, changes under systematic variations, and observed data counts at the same fidelity as used in the experiment. The pyhf JSON format describes the HistFactory family of statistical models [27], which is used by the majority of ATLAS searches. The pyhf package [28] is then used to construct statistical models, and perform statistical inference, within a Python environment. So far this is available for two analyses, the ATLAS 2019 sbottom multi-bottom search [29] and the search for direct stau production [30], both for full Run 2 luminosity. The provision of this full likelihood information is much appreciated and we hope that it will become a standard, as it greatly improves the quality of any reinterpretation.

An alternative method for publishing (almost) full likelihoods is to publish a machine-learning proxy for the likelihood function, trained by the experimental collaborations on the true likelihood function itself. Compared to full likelihood release, this approach has the obvious drawback that it requires a careful selection of proxy and training data to ensure that all relevant parts of the likelihood function, across the entire range of input parameters (whether kinematic variables or BSM theory parameters), are represented by the proxy function to sufficient accuracy. It however has the advantage of not requiring the release of as much proprietary experimental data. Such an approach has been proposed in Ref. [31], using a parameterisation that can encode complicated likelihoods with minimal loss of accuracy, in a lightweight, standard, and framework-independent format (e.g. ONNX) suitable for a wide range of reinterpretation applications. Applications to real data analyses are under discussion within the CMS collaboration.

### 2.1.5 Simplified model results

Interpretations in the context of simplified models have been pursued for many years by both ATLAS and CMS, in particular for SUSY and dark matter (DM) searches, and more recently for an increasing variety of LLP searches. Usually, limits are given in 2D planes of the simplified model parameters – often, but not exclusively, mother and daughter sparticle masses for SUSY searches, DM and mediator masses for DM searches, or LLP mass and lifetime in some LLP searches. Limits from resonance searches, commonly presented as bounds on the production cross-section times branching ratio as a function of the resonance mass for different assumptions on the width and the spin of the resonance, also fall into this class. This also includes limits from searches for BSM Higgs bosons.

We recall here, that 95% CL exclusion lines in 2D parameter planes are *not* useful for reinterpretation. More useful are 95% CL upper limits on production cross-sections or cross-section times branching ratio ($\sigma \times$ BR), simply referred to as "upper limits" in the following. Some searches also provide signal acceptances and efficiencies (or the product thereof) in the

same 2D planes. Such "efficiency maps" are particularly useful for reinterpretation, as they allow the combination of contributions from several simplified models [32, 33].

In a few rare cases (e.g. the CMS stop search [34]) the efficiency maps for a set of simplified models and for all signal regions are given in addition to a correlation matrix (see above). This information allows for the reconstruction of an approximate likelihood for the given simplified models and a more precise calculation of the analysis sensitivity for new models. Both upper limits and efficiencies (including correlation matrices) have been successfully used to reinterpret the experimental results, as has been demonstrated by the SModelS [33, 35, 36] program and its applications. It would be beneficial if simplified model efficiency maps were systematically provided for *all* signal regions, together with a (simplified or full) likelihood model for combining them.[4]

We also note some problematic issues in the reporting of simplified model limits. For instance, there are many searches which present results for particular combinations of final states (e.g. summed over lepton flavour, or assuming mixed decays with fixed BRs) or topologies (e.g. gluino pair production combined with gluino-squark associated production). In these cases, upper limits tend to be more model dependent, since the relative contribution of each topology or final state is fixed and cannot be varied for reinterpretation. Another issue arises if upper limits are given in terms of signal strengths instead of absolute cross-sections, as is often done in DM searches: this introduces a systematic uncertainty for reinterpretation unless the theory expectation used for normalisation is also given, with the same binning as for the signal strengths. Differences that result from kinematic distributions, and thus cut acceptances, and not just from the total signal cross-section, remain somewhat hidden in this case. An example of good practice is [37], which published maps of reference cross-sections as auxiliary material. Alternatively, signal cross-sections agreed upon, e.g. by working groups such as the LHC DMWG, could be put in a common, versioned, citable online repository (e.g. on Zenodo) and referenced by all.

In order to enable a systematic and powerful reuse of simplified model results, we hence give the following recommendations:

1. Simplified model topologies should aim to be as unbiased as possible by an underlying UV model, even when a specific model is used to generate the signal samples. In particular, individual results should be provided for each topology and final state. As an example, consider pair production of gluinos, each of which can decay to $b\bar{b}$ or $t\bar{t}$ plus the lightest neutralino. In this case we propose that efficiency maps be provided for the $4b$, $4t$, and $2t2b + E_T^{\mathrm{miss}}$ final states separately rather than their mixture resulting from fixed branching ratios. We stress that only with this information can one apply the experimental results to arbitrary models.

2. For a higher-dimensional parameter space (three or more mass parameters), occurring e.g. in cascade decays with more than one step, a full exploration of the parameter space is sometimes not feasible and, hence, fixed mass relations for intermediate particles in cascades are used. We suggest here to provide at least three values for each of the respective mass relations, in order to assess the dependence of the analysis' sensitivity on these parameters. In the case of LLP searches, it is also important to present results for distinct LLP lifetime values, since they strongly affect the signal efficiency. Generally, for the auxiliary material it would be preferable if efficiencies were released in a format that goes beyond the two-dimensional parameterisation suitable for paper figures whenever necessary – we suggest multidimensional data tables instead of a proliferation of two-dimensional projections of the parameter space.

---

[4]If this is not feasible because of the sheer number of signal regions, as is the case for some SUSY searches, appropriately aggregated (super-)signal regions have proven useful.

3. We recommend that efficiency maps be provided *for all* signal regions (or appropriately aggregated signal regions). This is relevant because the sensitivity of specific regions may change for different signal models. In addition, appropriate (simplified or full) likelihood information should be given to allow for the combination of signal regions, cf. the dedicated discussions in Sections 2.1.2 and 2.1.4.

4. For upper limits, it is useful to report both the observed *and* the expected limits as functions of the simplified model parameters, as this allows for selecting the most sensitive result and/or for computing an approximate likelihood as a truncated Gaussian [38]. If results are given in terms of signal strength (i.e. normalised to a theory expectation) instead of absolute total cross-section, the reference cross-sections should be provided in addition.

5. The presentation of results for various simplified models can significantly enhance the (re)applicability of the search. Since distinct topologies and final states can drastically change signal efficiencies, it is desirable to derive results for multiple simplified models for a given search.

6. Reference signal cross-sections used to normalise upper limits such as in DM searches or to obtain the exclusion curves should be provided as auxiliary material on a versioned, citable online repository.

It is clear that the above recommendations cannot be fully applied to all possible experimental searches. In particular, more sophisticated analyses beyond simple cut-and-count cannot always provide results in the form of efficiency maps. Furthermore, some of the simplified models considered do not allow for the factorisation of topologies, due to relevant interference effects between apparently empirically distinct topologies, as occurs in some mono-*X* searches. Providing efficiency maps or upper limits for multiple simplified models as recommended above might be impractical in cases. Nonetheless, providing results for at least two sufficiently distinct simplified models is still necessary so the dependence on the accessible kinematic phase space is apparent. This is particularly relevant for LLP searches, where e.g. the LLP displacement or anomalous ionisation/timing can be strongly dependent on the kinematics of the process and, hence, on the topology considered [39]. The above recommendations are not only important for the wider reuse of the respective simplified model results, but also provide decisive information for the validation of any analysis recasts.

Good practice of the provision of simplified model results in digital form in synchronicity with the release of an analysis was upheld by the CMS SUSY group searches during Run 1. The community would like to encourage continued systematic provision of this sort, as well as the continual efforts made by ATLAS to increase the amount of results provided on HEPData.

### 2.1.6 Statistical method

Although a given data set can of course be interpreted using many different statistical methods, it is desirable to be able to implement the same statistical procedure used in the analysis for the purpose of validation. All information necessary for this purpose should be provided, in particular regarding the treatment of nuisance parameters as well as the interpretation of signal regions with small numbers of events. For example, we note that there are several different definitions of the test statistic used to calculate $CL_s$ values [40]. Whenever exclusion limits are constructed by simulating pseudo-experiments, it would be useful to quantify how the expected limit would differ in the asymptotic limit. Whenever a Bayesian analysis is performed, prior distributions employed for all parameters entering the analysis should be fully specified.

### 2.1.7 Further metadata

Prime examples of this catch-all final category are input (SLHA) files and cut-flow tables for benchmark theory points. Together, these allow a detailed validation of any reinterpretation code, by confirming that it accurately reproduces the main features of the experimental analysis. Indeed, since the code repositories of ATLAS and CMS have been made public, this information or some approximation to it can sometimes be located, but direct linking to it from analysis records and making the presentation more uniform and systematic would greatly assist reinterpretation work.

**Model data and event generator:** model data are sometimes available, but not always and often in a very minimal form. More completeness of this information would greatly assist re-use. Concretely, we ask that

– model input files, a.k.a. SLHA files, used for the signal generation be provided (not only for SUSY but for all new physics searches) if not available already in a common, properly referenced repository;

– if an SLHA file is to be used as a template for a parameter grid, it helps to provide some annotations in the file and/or a brief usage description in a readme file;

– all the benchmark scenarios used in the paper are covered by the supplied model files; and

– that MC-steering information be provided in addition to the SLHA file(s); this includes the generator version number, generator run cards and, importantly, the relevant model implementation (e.g. the UFO files).

**MC samples:** As the matrix-element and parton-shower MC codes are public and not written by the experiments, and in this document we have already advocated publication of full generator run cards, the obvious next step is direct sharing of truth-level MC samples generated from these components. Indeed, this approach was taken earlier in the LHC programme for parton-level "LHE" MC event samples, for instance in the context of the Run 1 CMS analyses, but has fallen out of favour. Unlike real data or reconstruction-level MC specific to each experiment, MC truth samples reflect more an investment of CPU time and debugging effort than key experimental innovations. Hence, without undervaluing the effort committed to such sample generation, their impact would in fact be larger if shared between the experiments and the phenomenology community, the exactly equivalent samples more easily enabling direct comparisons and benchmarkings of analyses and interpretations. Such sharing would also minimise duplications of very substantial CPU expenditure for large samples of precision-calculated SM background processes, as we enter an era of collider physics in which MC computing budgets can be a limiting factor for many analyses.

**Cut-flows:** The presentation of cut-flows, i.e. numbers or fractions of events surviving sequential event-selection cuts, would benefit from standardisation. Frequent issues include cut orderings not matching the text (e.g. mixing up pre-selection and specific signal-region cuts), and several different normalisation schemes including fixed numbers of input events, fixed cross-sections, and cut-flows expressed as chains of one-step efficiencies. We suggest that the information be presented primarily as a cumulative percentage or fraction of events passing, with a sufficiently high precision. If full MC run information is provided, the normalisation to

the MC cross-section could be useful additional information, but as this just constitutes a single scale-factor the same effect can be achieved more compactly by presentation of expected yields for all signal regions. Indeed this usually provides *extra* information, as cut-flows are often reported only for the cuts common to all signal regions. Cut-flows are useful before the analysis is final, and should be relatively uncontroversial, being only MC information. We note that both the ATLAS and CMS full Run 2 zero-lepton SUSY searches [41,42] have indeed provided cut-flows early on; the CMS cut-flow [42] and its 36 fb$^{-1}$ iteration [43] are particularly apt templates for good practice.

### 2.1.8 Pseudocode, code snippets

There is another way to ensure the correct operation of analysis-recasting programs: for the experiments to publish their own, validated versions. This is now regularly attempted by ATLAS SUSY analyses as additional resources in HEPData submissions, effectively in pseudocode from the perspective of external recasters [24,29,30,44–47]. This is a highly welcome development, that has proven extremely helpful in some cases, and we greatly encourage that it be pursued further. Nonetheless there are some limitations due to the non-executability of the provided example code: we encourage experiments to publish both executable analysis logic and, if not already public, the framework in which they are to be run.

Analysis logic has become semi-regularly published for measurement analyses, dominantly as analysis codes to be used in the Rivet framework [48,49]: we welcome this development but note that coverage is nonetheless patchy [50], including in areas such as top-quark and Higgs physics where significant BSM sensitivity may be obtained through global fits. We hence recommend that experimental collaborations make provision of measurement Rivet routines a required step in publication, for relevant analyses.

For searches, pseudocode implementations are a welcome step, but they are still relatively rare and cannot be compiled and therefore tested directly. Significant errors have also been discovered in the logic of a number of these implementations during recasting exercises, reflecting the fact that these codes were not used directly in the analysis: had they been, the issue would have been immediately obvious. Workflows that allow real code from the data analysis to be usefully made public are the ideal form to guarantee accuracy: indeed, ATLAS' SimpleAnalysis internal framework is often the real, executable context for provided code snippets. But it is also important that the analysis code framework itself be public, as (for usability by experimental analysers) complex operations like lepton isolation or overlap removal between physics-objects are encapsulated in opaque functions (e.g. SimpleAnalysis' `overlapRemoval()`), whose signature alone does not help recasters with implementing equivalent logic. As offline experimental code is already public, publishing experiment analysis framework code should not introduce any experimental data-security issues. We hence encourage experiments to publish more executable analysis logic, and the minimal analysis framework that they run in.

Another way of describing the analysis logic is to use an analysis description language [51, 52], a domain specific, declarative language designed to express the physics contents of an analysis in a standard and unambiguous way. In this approach, description of the analysis components is decoupled from analysis frameworks, but the analysis can be run by any framework capable of interpreting the language, which makes the approach commonly usable by experimentalists and phenomenologists. Considerable progress has been made in developing analysis description languages and interpreter frameworks [20,53,54]. We also note nascent efforts [52] to develop a single community framework for BSM collider analysis recasting.

Pseudocode (or declarative-language descriptions) however cannot be easily provided in the ever-growing number of cases where ML methods are used in searches. These require the computation of the specific distributions of many, sometimes hundreds, of high- and low-

level analysis quantities. This is an open problem, discussed to some extent in section 2.1.10. Generally, we note that these new techniques pose new challenges for reinterpretation, and community-wide dialogue is needed to address them effectively.

### 2.1.9 Direct analysis code preservation

A complement to the discussion so far, which has concerned provision of information needed to make approximate reproductions of fully detailed experimental analyses, is the concept of fully preserving an analysis' computational toolchain such that an *exact* repetition is possible. Particularly notable in this domain is the Reana/Recast [55] system, which uses Docker containers and workflow description languages to encapsulate both the orchestration of data flow and processing tools, and the original experiment software platform on which the analysis was performed. New Run 2 BSM search analyses in ATLAS are now required to implement Recast preservation, a procedural measure key to achieving comprehensive analysis coverage.

This development is a major step forward for full reproducibility of LHC analyses, but with the accompanying CPU cost of full-detail experimental data processing. The evolution of computing power and the efforts of experimental collaborations to provide fast simulations that produce the required input for the preserved analysis will ameliorate this cost. While the prospective users of full analysis preservation are for now the LHC experiments themselves, use of full-detail preservation on models targeted by lightweight reinterpretation fits may become feasible for phenomenologists in the future. We hence recommend the public availability of analysis Recast images, perhaps after an embargo period (cf. Open Data), and a combined preservation strategy mixing lightweight and full-detail frameworks. In the meanwhile, it might be interesting to tag analyses, for which Recast images exist in HEPData so that interested phenomenologists could know and eventually propose reinterpretations to be run.

### 2.1.10 Open questions

There also exist some unresolved issues. One is the reinterpretation of prompt searches in the presence of both prompt and displaced objects. First, it is often unclear how jets or leptons from decays of LLPs are seen in ordinary prompt searches, especially when they accompany prompt objects. Are they ordinary jets/leptons? Are they just calorimeter deposits contributing to missing energy calculation? Are they simply removed as part of cleaning? Providing a lifetime window where objects can be considered prompt will partly help to answer this question by allowing one to determine the fraction of events from LLP decays with entirely prompt signatures. However, we also lack information about how events that satisfy all prompt criteria but contain additional displaced objects, are to be treated. As such, we do not have an agreed-upon, tested mechanism for reinterpreting the reach of prompt searches for models that predict events with both prompt and displaced objects. A study of the overlap of long-lived and prompt searches has been performed by ATLAS [56, 57]. This study shows that jet-based searches include "cleaning cuts" that would drastically remove products from LLPs while leaving products of prompt decays unaffected. It is as yet completely unknown how such cleaning cuts could be modeled and whether a simple factor proportional to the lifetime would be sufficient.

Another completely open question is, as already mentioned, the reinterpretation of analyses which employ ML techniques for the signal/background separation. Such analyses currently cannot be reproduced at all outside the experimental collaboration. Generally speaking, a ML algorithm learns to predict an output variable $y$ (or a set of output variables $\vec{y}$) given a set of input variables $\vec{x}$ via building an estimator of the mapping function $\hat{f}(x) = y$. In principle the resulting model, that is the weights defining $\hat{f}(x) = y$, can be stored together

with appropriate meta data (describing the inputs and outputs of the model, as well as other relevant information like e.g. the boundaries of the training region), thus allowing external users to reuse this model for their own input. Depending on which library was used to create the trained model, different formats may be suitable. For scikit-learn, for example, one may serialise the entire model object with the pickle package in Python. Another option, in particular for neural networks, would be to store the model in an ONNX file [31]. The publication of trained ML models for HEP phenomenology is discussed in detail in the contribution by S. Caron *et al.* in Ref. [52]. In the context of LHC experimental analyses, one of the difficulties is the number and type of input variables and nuisance parameters—for reuse outside the collaboration, in many cases a simplified ML model would need to be constructed, based on only the most relevant input variables, provided they can be generated in a simulation. Other questions concern the mapping between reconstruction and truth levels, technical robustness, etc. The challenges are certainly manifold, but we strongly encourage pioneering feasibility studies.

## 2.2 Measurements

By "measurements", we primarily mean cross-section measurements and flavour physics observables. These are defined in terms of the final state and can have a high degree of model independence. Measurements of SM parameters (for example the $W$ boson or top masses) are made in the context of the SM and are most easily interpretable as consistency checks within that context. We do not discuss them further here. Other types of measurements, relevant in particular for the Higgs sector, e.g. pseudo-observables, signal strengths, and simplified template cross-sections, are discussed in detail in Refs. [58, 59]. They typically carry additional model dependence due to assumptions about production processes and kinematic templates, but can be very powerful in some contexts. We return to them briefly in subsection 2.2.7. Measurements from Open Data are discussed in Section 2.3.

Cross-section measurements have a history of use in tuning non-perturbative parameters of MCEGs, validating the implementation of the SM in MCEGs, and as input to parton density function (PDF) fits. The information provided typically enables and reflects these use cases. However, measurements are also increasingly used to extract SM parameters and to constrain BSM physics, either in effective field theory (EFT) frameworks [60, 61] or more recently by direct comparison to BSM final states [62]. Flavour physics observables require treatment less often compared to cross-section measurements due to low background and high resolution for fully reconstructed final states. Exceptions to this rule are discussed below. Measurement analyses typically provide the following information:

- Primary data: typically binned, unfolded observable values. As with searches, this is available in the published papers, and increasingly available electronically;

- Background estimates: usually not provided and not needed, but see below;

- Correlations: increasingly provided as standard in measurements;

- Likelihood: in some cases where the likelihood is not Gaussian (parabolic), the full likelihood (log-likelihood) of a measurement is provided;

- Reproduction metadata: the precise definition of a "final state particle" and of the fiducial kinematic region of the cross-section is essential and is usually provided;

- Theoretical predictions: measurements can be compared to precision SM calculations, as is usually done in experimental papers. However, such predictions are only rarely provided in digital form.

As with the previous section, we now discuss each of these items in turn, with suggestions for facilitating reinterpretation and for securing the long-term use and legacy of the measurements.

### 2.2.1  Primary data

Differential fiducial cross-sections are usually presented after unfolding to the final-state particle level. Additional theory correction factors (for effects such as electroweak final-state radiation, or hadronisation) are sometimes also provided to ease comparison to different levels of theory prediction.

For flavour physics observables, unfolding is not usually necessary unless the final states involve neutrinos. Such semi-leptonic decay measurements are often differential, and a response matrix is provided. A good example of this is Ref. [63] which was reinterpreted in Ref. [64]. Given the statistical ambiguities associated with unfolding, there is also a drive towards providing the raw distributions with the response matrix that is used to fold the probability density function such as in Ref. [65], rather than to present unfolded data. In all analyses it is recommended to at least internally determine the results with both folded and unfolded approaches to check systematic uncertainties.

For distributions sensitive to BSM contributions in high-energy tails, information about the highest data point should be publicly released as auxiliary information alongside any measurement, as this provides a key input to unitarisation procedures in reinterpretation. We also recommend that any unitarity constraints on models tested be made explicit and that any BSM fits be conducted both with and without them to test the applicability of any derived limits.

We recommend that tests of the measurement procedure be conducted within experimental collaborations for a range of BSM/SM injected hypotheses, and for publications to present any implicit assumptions or limitations on the validity of published data. Such biases can potentially arise particularly through treatment of background subtraction/modelling (where background rates are affected by the same BSM physics as the signal of primary interest), and in correction for detector response. Such effects can be mitigated but require careful study.

### 2.2.2  Background estimates

Generally backgrounds are subtracted before or during the unfolding procedure, and the uncertainties are included in the measurement uncertainties. For instrumental backgrounds, this is a consistent and functional approach and no further work is needed for reinterpretation. For irreducible backgrounds, i.e. processes other than those explicitly targeted by the measurement that nevertheless contribute to the fiducial phase space, this may build in an assumption that the SM is correct, making reinterpretation potentially problematic. From a reinterpretation point of view, defining the cross-section solely in terms of the final state is highly desirable; that is, performing no subtraction of irreducible backgrounds. Because this complicates SM interpretation, such subtraction may be done at particle level (see for example Ref. [66]) and indeed can then be redone independently, should the background predictions be improved at a later stage. We also recommend that experimental collaborations release the SM predictions associated with each measurement (with uncertainties).

Lepton universality measurements of the type $b \to c \ell \nu$ [67–69] suffer from large backgrounds, rendering subtraction unfeasible. In addition, the signal determination is performed using kinematic variables whose distributions are sensitive to BSM physics. This significantly complicates the interpretation of such measurements, where currently SM decay kinematics are assumed. This inconsistency can be resolved by the HAMMER tool [70], which can be

used to efficiently and consistently vary the kinematic distributions in order to provide measurements of the underlying Wilson coefficients. Such measurements could thus be used in a more general context than the lepton universality ratios currently provided.

### 2.2.3   Correlations

A breakdown of systematic uncertainties into several correlated components is increasingly made available in HEPData, and is already used in SM applications such as PDF fitting and global fits to flavour observables. To maximise the use of measurements for interpretations, we recommend that in addition to publication of measurements and uncertainties in HEP-Data, statistical covariance matrices and signed systematic shifts by uncertainty source are released in a systematic manner. Such information is also crucial to enabling combination of observables and datasets in global fits.

LHCb has been a pioneer in providing correlation matrices (e.g. [25]). More standardisation, as already discussed with regard to searches, would improve efficiency and reduce the chance of error. For technical considerations and recommendations regarding correlations we refer again to Appendix A.

### 2.2.4   Likelihoods

Interpretation of measurements that have non-Gaussian uncertainties requires information about asymmetries, for example, at least the diagonal skew matrix (see Ref. [13]). This is normally the case when the measurement is based on either a small dataset or is close to physical boundaries. This is particularly important when probing the large statistical significances which are far away from the central value of the measurements. For example, the LHCb collaboration published the full likelihood of the $R_{K^*}$ measurement Ref. [71] because of the small data set it was based upon.

This approach is viable for both searches and measurements. However, the large complexity of an experimental analysis, where $\mathcal{O}(100-1000)$ nuisance parameters are employed to describe the dependence of measurements on a wide variety of theoretical and experimental systematic uncertainties often render the full experimental likelihood unsuitable for phenomenological reinterpretation. A simplified approach based on the JSON format [11] has been proposed and might offer a way where only the most relevant theoretical (and, if necessary, experimental) systematic uncertainties are retained separately, and all other uncertainties are combined. We would like to encourage the phenomenological and experimental community to engage in tests of this approach.

### 2.2.5   Desirable reproduction metadata

In order to reproduce a measurement, our main requirement is the precise definition of a "final state particle" and of the fiducial kinematic region of the cross-section measurement. This is usually provided in the paper, sometimes supplemented by a Rivet [48, 49] implementation. We recommend that providing a Rivet routine becomes the standard for every applicable published measurement. For reproduction of MCEG curves in the paper, generator versions and key parameters are usually also given; providing them in addition in terms of runnable configuration files would again improve efficiency and reduce errors.

### 2.2.6 Theoretical predictions

These are either provided directly by theorists, or more usually produced by the experiments running open-source codes provided by (and with the guidance of) theorists. For cross-sections, given a particle-level cross-section definition, they can in principle be regenerated by anyone. However, this typically requires significant expertise and processing time. Some analyses do now provide the theory predictions using the HEPData multiple-axis capability, and expanding this provision such that the SM predictions are readily available would greatly aid BSM reinterpretation.

In the case of EFT limits it is recommended that a "clipping scan" is performed in which BSM signal contributions are set to zero for $\sqrt{\hat{s}} > E_c$, where $E_c$ is a cutoff scale [72] that is a free parameter. This enables assessment of the dependence of limits on unitarity bounds of the model, and the results of these studies should be published and any and all modifications made to original BSM models to account for unitarity constraints should be publicly documented to allow for proper reinterpretation.

In many rare decay measurements, the theoretical uncertainties are of comparable size or larger than the combined experimental uncertainties. Constraining these uncertainties by fitting theoretical models to the raw data has been proposed [73–75]. However, even with full correlations of theoretical nuisance parameters, this anchors any (re-)interpretation to a particular theoretical model. Changing the theoretical parametrisation requires a refit to the raw data which is often not provided (as it can be seen to be too sensitive to release). The solution to this issue will require close collaboration between the theoretical and experimental communities.

### 2.2.7 Higgs signal strengths and STXS

Simplified Template Cross-Sections [59] (STXSs) provide a staged approach to measure cross-sections of Higgs boson production categorised by production mode and by bins of particle level kinematics. The separation by production mode sets it apart from fiducial differential cross-sections, however it introduces model dependence. The staged approach allows to adapt the level of kinematic complexity and of separation between production modes to the experimental sensitivity, which increases with growing luminosity and differs for the production modes. The STXS framework is suitable for easy phenomenological interpretation because the observables are defined at the MC particle level and can be interpreted for separate production modes without detector simulation. It has proven highly useful in particular in the EFT context, where higher-dimensional operators may affect the kinematics of Higgs-boson production.

The model dependence arises due to the underlying MC simulation (which assumes a *SM-like* Higgs boson) that is employed in the transition from the experimental measurements to the particle level interpretation separated by production mode. Given the widespread use of the STXS framework, we propose a model-independent addition based on the experimental reconstructed event categories in the STXS framework: Publishing full visible cross-sections or signal strength multiplier results in these experimental categories, together with all relevant uncertainties, relative channel efficiencies and reference results, as outlined below, would help alleviate the model dependence.

In addition, the signal strength framework remains of high interest because of its power and ease of use in constraining new physics which has the same tensor structure as the SM: a multitude of theory papers and two widely-used public codes, HiggsSignals [76] and Lilith [77,78], are based on Higgs signal strength results. We therefore encourage the experimental collaborations to continue to provide detailed Higgs signal strength multiplier results ("$\mu$ values") in addition to STXS and differential fiducial cross-sections. For optimal usefulness, we would

appreciate:

- best-fit $\mu$ values and uncertainties for the experimental reconstructed event categories *and* for the pure, i.e. unfolded, Higgs production × decay modes (allowing for negative $\mu$ values in order to have an unbiased estimator);

- channel-by-channel correlation or covariance matrices, separately for experimental and theoretical uncertainties, as well as for the total (i.e. combined) uncertainties. This result should be provided for both the reconstructed event categories and the unfolded production modes;

- reference values (normalisation) of the corresponding SM predictions, or a clear reference where these values can be found, in the same binning as the experimental results;

- signal efficiencies per signal channel assuming SM Higgs kinematics; moreover, when results are given for a combination of production and/or decay modes, their relative contributions wherever possible.

Generally, signal strength measurements for a fixed (best-fit) Higgs mass are preferable over measurements where the Higgs mass is profiled. If a strong dependence of the signal strength result on the mass is observed, a two dimensional presentation of the main results could be provided in addition. Example fits of reduced couplings are highly useful for cross-checks and validation. We also note that it is crucial for a robust usage of the experimental results, that the numbers quoted in tables and on figures be precise enough to accurately reproduce the official coupling fits (cf. the discussion in Section 5.1 of Ref. [78]).

### 2.2.8 Summary of key recommendations

To summarise, our recommendations can be divided in two main topics. First, we advise that flavour-physics measurements:

1. Should be determined with both unfolded and folded data.

2. Sensitivity to underlying kinematics should be evaluated and published.

Second, regarding differential cross-section measurements, we recommend that:

1. Information about the highest data point in the tails of distributions should be made available.

2. Any implicit limitations on the validity of unfolded fiducial cross-section definitions should be assessed and made explicit, and where possible they should be tested by BSM signal injection.

3. Fiducial cross-sections should be defined solely in terms of final-state particles, without subtraction of irreducible backgrounds.

4. Uncertainty correlations, including statistical correlations, should be made available and broken down by source.

5. The SM predictions should be made available in HEPData along with the measurement.

6. A Rivet routine should be provided.

7. Visible cross-sections for STXS categories should be published.

For recommendations specific to Higgs signal strengths, see the items in 2.2.7.

## 2.3 Open Data

The term "Open Data" refers to both the actual collision datasets recorded with the LHC experiments as well as the corresponding simulated datasets. Only with both ingredients available can physics analysis results be fully reproduced. Furthermore, this also enables the possibility of producing new results. The main requirements for providing user-friendly open data is a large amount of disk space and the person-power needed to make the data technically available in a user friendly way. These requirements pose a significant challenge for some experiments.

Open Data are made available via the CERN Open Data Portal [6]. By far the largest set of Open Data, amounting to more than 2 Petabytes, is provided by the CMS Collaboration. Based on the collaboration's data preservation, re-use, and open access policy [7], 50% of the collision data are published a few years after data taking, and up to 100% within ten years. The LHCb Collaboration has made available a set $D \rightarrow K\pi$ candidates for outreach purposes and avenues on how to expand this are currently being pursued. Open Data initiatives from the ATLAS Collaboration currently have a primarily educational focus [79] but future data releases and policy could be designed to enable scientific research and outputs.

Data produced by the LHC experiments are usually categorised in four different levels [80], ranging from open access publication of the results to allowing the full reconstruction of raw-level data, providing the software to reconstruct and analyse them. Over the course of the last few years, the CMS Collaboration has even made this last level available. The data have been used extensively for outreach and education, and the list of actual physics publications is constantly growing, see e.g. Refs. [81–86]. Currently, 100% of the 2010 and 50% of the 2011, and 2012 collision data from CMS are publicly available.

Analysing the experimental data with a view to providing the necessary information for the simplified approaches discussed in the sections above is challenging. Extensive documentation is required to understand the physics objects as well as how to use the software. Therefore, work is ongoing to provide further analysis examples that demonstrate how one can obtain scientific results. Furthermore, a simplified data format is under development that will allow analysis without experiment-specific software. We strongly recommend theorists and other interested parties to make use of the Open Data available, and to ask for help and clarification if needed.

Additional challenges are posed by the computational environment and the overall computational effort required to process the data. In general, the operating systems and architectures used in past data-taking will no longer be available and maintained at the time of reinterpretation. Therefore, virtual machines and software containers have been made available that can be run on modern computing platforms. These also allow the execution of analysis jobs on high performance computing platforms that can be leveraged on demand by renting them from a cloud provider.

Open Data initiatives from the ATLAS Collaboration are currently directed to educational and public engagement use. The available datasets correspond to 10 fb$^{-1}$ of newly-released 13 TeV data [87] (in ROOT format) containing events with either one or more leptons, hadronic jets, hadronic taus, or high-energy photons, and also 2 fb$^{-1}$ (XML format) and 1 fb$^{-1}$ (ROOT format) of 8 TeV data for events with at least one electron or muon. Also released are associated MC simulations of SM processes and some BSM signatures. These data sets are designed to be used without the need for specialised software and provide a compelling model for the future use of ATLAS Open Data for scientific purposes.

Future releases of 13 TeV data sets from ATLAS would provide an excellent opportunity for ATLAS to expand the information content in released data sets to enable scientific studies by the wider community. We recommend that ATLAS consider the development of a policy for release of data sets after an appropriate time delay (of a few years). Such data (and

associated MC simulations) should include the full range of trigger and final state particle signatures available, building on the initiative of recent 13 TeV data releases. To maximise the scientific use of the data we recommend these open data sets include not only four-vector information on final state particles and associated detector-level quantities (isolation, flavour-tagging information, quality criteria) but also incorporate systematic variations associated with measurement uncertainties.

As demonstrated in Ref. [83], which uses CMS Open Data, open collider data has the potential to assist the BSM search programme at the LHC. First, Open Data can be used to study questions that are outside the mainstream search programme, and thus explore new territory. Second, when important backgrounds are challenging for theorists to simulate reliably, Open Data can provide those backgrounds directly, making phenomenological studies or prototype analyses far more accurate. Although searches using current Open Data releases are unlikely to uncover BSM phenomena on their own, they can help demonstrate the value of certain search strategies and justify the application of those strategies by the experimental collaborations on much larger data sets.

## 3 Comparison of reinterpretation methods

The reinterpretation of experimental results can generically be done in two ways: by applying appropriate simplified model results to more complete models, or by reproducing the experimental analysis in a MC simulation (aspects that concern specifically flavour physics will be discussed later).

The simplified-model approach assumes that the signal selection is sufficiently inclusive so that possible differences in kinematic distributions (between the original simplified model and the new model or scenario) do not significantly impact the acceptances. All one then needs to compute is signal weights in terms of cross-sections × branching ratios (× efficiencies). There is one current public tool, SModelS [33, 35, 36], which applies this approach generically to BSM models. Clearly, this is easier and much faster than event simulation and additionally it allows for reinterpreting searches which are not cut and count, e.g. analyses which rely on BDT (boosted decision tree) variables. The downside is that the applicability is limited by the simplified-model results available. Moreover, whenever the tested signal splits up into many different channels, as typically the case in complex models with several new particles, the derived limits tend to be highly conservative and often far too weak [88,89]. Tools which evaluate cross-section × branching ratio limits for specific models/signatures, such as HiggsBounds [90,91] for additional Higgs bosons, ZPEED [92] for $Z'$ resonances, and Dark-Cast [93] for dark photons, also fall into this class.[5]

Reinterpretation by means of MC simulation is more generally applicable and more precise but also more difficult and much more time consuming. There are a number of public software frameworks with this aim, each coming with their own set of implemented analyses: CheckMATE [95,96], MadAnalysis 5 [18,19,97,98], GAMBIT's ColliderBit [12,99–101] and Rivet [48,49] (and hence Contur [61], which interprets Rivet analysis outputs). There are also two interpreters of the recently developed domain-specific Analysis Description Language (ADL) [51,52]: adl2tnm [20] and CutLang [53,54]. For LLP searches, there is no standard framework yet, but a number of recast codes have been made available via the "LLP Recasting Repository".[6] A noteworthy aspect is that, in a simulation, searches and measurements can in

---

[5]Two more such codes, fastlim [32] for the Minimal Supersymmetric Standard Model (MSSM) and XQ-CAT [94] for heavy quarks, exist but have not been updated since (early) Run 1.

[6]Long-lived particle searches rely on non-standard, detector-dependent definitions of reconstructed objects. As a result, publicly available fast detector simulation codes like Delphes currently cannot be used without significant

principle be treated on the same footing; indeed MadAnalysis 5 and in particular Rivet include both types of analyses.

Regarding searches, in all the above tools it is so far always assumed that no new backgrounds need to be considered and the hypothesised signal does not affect control regions. One can then simply determine the event counts in the signal regions and compare them to the 95% CL observed limits, or take the numbers of observed events and expected backgrounds to compute a likelihood. One major difference lies however in the emulation of detector effects, which is necessary as searches are typically not unfolded. Tables 1 and 2 summarise the available frameworks for reinterpretation and compare the strategies used by each of them. Further details of each tool can be found in the following subsections. Direct comparisons of the performances have been started [20, 52] but without definite conclusions so far.

Machine learning can also help in the process of generalising likelihoods or exclusion boundaries. Model exclusion boundaries or likelihood (ratios) can be learned, explored and provided for further use using ML models, see e.g. [102, 103]. The models are made by learning numbers based on experimental measurements (e.g. likelihoods, posteriors, confidence levels for exclusion) from training data given the parameters of the physical model and experimental nuisance parameters [31]. Such training data can come from experiments or the recasting tools discussed below. This topic is discussed in some detail in the ML contributions in [52].

## 3.1 Public tools for interpretation of BSM searches

In GAMBIT [99, 100] the reinterpretation of collider results is handled by the ColliderBit module [101]. This currently allows for the fast simulation of LHC events using a parallelised version of Pythia 8 [105, 106]. A large range of BSM searches from ATLAS and CMS are implemented (26 from Run 2, 12 from Run 1), with a focus on searches for supersymmetry. Emphasising the combination of full event generation and speed, ColliderBit uses a smearing approach to detector simulation, through its BuckFast routines, and takes published efficiency functions from the individual experiments. Cross-sections are currently taken from Pythia; future versions will support cross-sections from other external tools. In order to supplement the existing BSM processes in Pythia, ColliderBit can make use of the interface between Pythia and MadGraph5_aMC@NLO [107], allowing the generation of matrix element code. A standalone tool built on ColliderBit (known as ColliderBit Solo; CBS) is also able to apply ColliderBit detector simulations and analyses to events provided in HepMC format by other MC generators.

The Rivet toolkit [48, 49] is established at the LHC as an analysis tool and receptacle for preservation of measurement-analysis logic. Since version 2.5.0 in 2016, it has also provided a sophisticated detector efficiency and kinematic smearing system for reconstruction-level BSM searches, built on top of the existing particle-level observable calculators for construction of particles, jets, and missing momentum. This design was developed based on experience with GAMBIT's ColliderBit and Delphes, and allows every analysis to bundle smearing and efficiency functions specific to its phase-space in the analysis code, if generic functions based on experiment performance notes are insufficient. Around 30 BSM analyses are currently bundled with Rivet. As of version 3.0.0, Rivet also transparently propagates MC systematic uncertainties via event weights. Rivet is not a comprehensive reinterpretation system: it does not itself provide machinery for scanning of parameter spaces or computation of likelihoods. Work is currently ongoing to interface Rivet to GAMBIT as a new supplier of inputs for computing collider likelihoods.

---

modifications to define the new objects.

Table 1: Summary of public frameworks for the reinterpretation of searches and measurements. The columns summarise the major inputs from the experiments used for the reinterpretation, how detector effects are modelled (if necessary) and the principle outputs in terms of performing statistical inference. Particle-level inputs specifically refer to files in HepMC format, whereas parton-level inputs specifically refer to LHE files (except in the case of Recast, which can also accept other internal ATLAS parton-level formats).

| Package | Refs. | Experimental inputs | Event input | Detector simulation | Inference/Output |
|---|---|---|---|---|---|
| GAMBIT (ColliderBit) | 12, 99–101 | Cut-flows, analysis logic, object-level efficiency functions, observed event numbers in signal regions, background covariance matrices | particle | BuckFast (smearing & efficiencies) | Detector-level distributions, signal region efficiencies, simplified likelihood for calculating exclusion limits/contours |
| CheckMATE | 95, 96 | Cut-flows, analysis logic, object-level efficiency functions, observed event numbers in signal regions | particle, parton | Delphes | Detector-level distributions, signal region efficiencies, ratio of predicted to excluded cross-section |
| MadAnalysis 5 | 17–19, 97, 98 | Cut-flows, analysis logic, object-level efficiency functions, observed event numbers in signal regions, background covariance matrices, JSON likelihoods | particle | Delphes; customisable smearing | Detector-level distributions, signal region efficiencies, $1 - CL_s$ values |
| Rivet | 48, 49 | Cut-flows, analysis logic, detector smearing & efficiency functions | particle | Customisable smearing | Truth/detector-level distributions |
| Contur | 61 | Unfolded (particle-level) differential cross-sections via Rivet | particle | N/A | Exclusion contours in BSM model space |
| ADL interpreters: adl2tnm, CutLang | 20, 53, 54 | analysis logic, external functions of complex variables, object or event level efficiencies | particle | External (Delphes, CMS and ATLAS simulations) | cutflows, event-by-event weights per region, histograms |
| Recast | 8 | Experiment-specific formats | parton | Experiment-owned (fast) simulation | $p$-values, upper limits, likelihood values |

CheckMATE [95, 96] is a tool that uses detector simulation output from Delphes [15] to implement cut-and-count searches for collider experiments. Currently, its codebase of validated ATLAS and CMS search analyses includes 2 analyses at 7 TeV, 34 at 8 TeV and 21 at 13 TeV (along with 23 more that are publicly available but only partially validated due to lack of publicly available validation information). Since the release of CheckMATE 2, users can now generate events on-the-fly using Pythia 8 or MadGraph5_aMC@NLO by providing standard parameter and run cards for these codes. Adding a new analysis is streamlined by CheckMATE's AnalysisManager [108] which sets up the database of observed and expected number of events in each signal region, the relevant Delphes card and a library of standardised definitions of reconstructed objects used by ATLAS or CMS that will be needed. With the next

Table 2: Summary of public frameworks for the reinterpretation of searches and measurements (continued). The columns summarise the major inputs from the experiments used for the reinterpretation, the model inputs, and the principle outputs in terms of performing statistical inference.

| Package | Refs. | Experimental inputs | Model input | Inference/Output |
|---|---|---|---|---|
| SModelS | 33, 35, 36 | Simplified-model cross-section upper limits and efficiency maps from SUSY searches, background covariance matrices | SLHA or LHE (any BSM model with $Z_2$-like symmetry) | Ratio of predicted to excluded cross-section, exclusion CL (if efficiency maps are available) |
| HiggsBounds | 90, 91 | Model independent (exp. and obs.) 95% CL upper limits and exclusion likelihoods from BSM Higgs searches | masses, widths, cross-sections and BRs (or effective couplings) of all Higgs bosons | Ratio of predicted to excluded cross-section, allowed/excluded at 95% CL, $\chi^2$ for specific searches |
| ZPEED | 92 | Observed event numbers in signal regions, background predictions, detector resolution and efficiencies | Model parameters | Likelihood values |
| DarkCast | 93 | Simplified-model production mechanism, cross-section upper limits or ratio map of observed to expected cross-sections for dark photon searches | couplings of new gauge bosons to the SM fermions | 95% CL exclusion limits on couplings |
| DarkEFT | 104 | 95% CL exclusion limits on dark sector searches and rare meson decay BRs | effective couplings for 4-fermion operators | 95% CL exclusion limits on the effective coupling |

upcoming release, CheckMATE 2 will include three LLP searches, viz. the CMS displaced leptons, ATLAS disappearing track, and the ATLAS displaced vertex search, using generator-level efficiencies provided by the respective analyses.

The MadAnalysis 5 package [97, 98] is a general framework for new physics phenomenology aiming to ease the design and the implementation of collider analyses. Relying on Delphes for simulating the response of the ATLAS and CMS detectors, MadAnalysis 5 can be used to automatically recast the results of various ATLAS and CMS analyses (currently 18 Run 1 and 16 Run 2 analyses, including one LLP search). All analyses are available in a Public Analysis Database (PAD) [18, 19], which can be automatically installed locally; details on the implementations are provided in dedicated validation notes. MadAnalysis 5 can moreover be fully integrated in the MadGraph5_aMC@NLO/Pythia framework, achieving thus a high level of automation from hard-scattering event generation to the recasting of the detector-level corresponding sample. In its v1.8 release, MadAnalysis 5 also allows one to use efficiency and smearing functions to parameterise the detector effects [17], similarly to what is done in Rivet and ColliderBit. The usage [52] of covariance matrices and/or full likelihood information is planned for v1.9.

adl2tnm [20] and CutLang [53, 54] parse and run analysis logic written in the form of the recently developed domain-specific ADL [51, 52]. In the plain-text ADL files, object, variable and event selection definitions are separated into blocks that follow a keyword-value structure, where keywords specify analysis concepts and operations. The syntax includes

mathematical and logical operations, comparison and optimisation operators, reducers, four-vector algebra and common HEP-specific functions (e.g. $\Delta\phi$, $\Delta R$). ADL files can refer to self-contained functions encapsulating variables with complex algorithms (e.g. $M_{T2}$, aplanarity) or non-analytic variables (e.g. efficiency tables, machine learning discriminators). The Python program adl2tnm writes C++ analysis code from ADL files. CutLang is a runtime ADL interpreter, able to operate directly on events without compilation. Both packages can run on a variety of event formats. A repository of LHC analyses implemented in the ADL format is available in [109]. A parser from ADL to Rivet is also being developed [20].

The LLP Recasting Repository is a repository on GitHub holding various (typically Pythia-based) example codes for recasting LLP searches. The repository folder structure is organised according to the type of LLP signature and the corresponding analysis and authors. A README file can be found inside each folder with the required dependencies and basic instructions on how to run the recasting codes. Ideally, a note on the recasting procedure and some validation figures are also provided. The repository is open for everybody, and new code submissions are highly encouraged.

The approach taken by SModelS [33, 35, 36] does not employ MC event generation.[7] Instead, a given BSM model is decomposed into simplified model spectrum (SMS) components, whose weights ($\sigma \times$BRs) are then matched against a database of LHC results. The benefit of this approach lies in its speed; it takes only a few seconds to confront a given theory with the entire database of results. Another advantage is that prompt and long-lived searches can be treated in essentially the same way [36]. These advantages come at the price of being conservative: because only a part of all occurring signatures is constrained by corresponding SMS results in the database, the resulting limits will always be less constraining than the ones obtained from fully recasting a given analysis. The current version, v1.2.2, ships with a database of results from more than 60 ATLAS and CMS analyses (from both, Run 1 and Run 2), including several results for LLPs. An improved treatment of LLP signatures is currently in preparation. Presently, SModelS can only constrain simplified model topologies having a two-branch structure, i.e. topologies originating from the pair production of BSM particles followed by their cascade decay. Resonance searches, for instance, are not included in the current version. However, the procedure can be generalised to arbitrary topologies, which will be part of future developments.

The limits from BSM Higgs searches are encoded in a relatively model independent way in the code HiggsBounds [90, 91]. The data consist of the observed and expected 95% CL exclusion bounds, as well as the observed and expected likelihoods for some channels (whenever available). The user provides the mass, width, cross-sections and branching ratios for each Higgs boson in the model under investigation. Alternatively, effective couplings can be provided. For each Higgs boson HiggsBounds determines the most sensitive channel from the *expected* exclusion bounds; only this channel is then tested against the *observed* limit. If any of the Higgs bosons is excluded, the parameter point is considered excluded. The $\chi^2$ value for a parameter point can also be evaluated based on all channels where likelihood information is available.

Regarding spin-1 resonance searches, the recently released code ZPEED [92] provides fast likelihoods and exclusion bounds from dilepton resonance searches for general $Z'$ models. This is achieved by combining analytical expressions for leading-order differential cross-sections with tabulated functions that account for PDF effects, phase space cuts, detector efficiencies, energy resolution and higher-order corrections. The same approach is taken for the Drell-Yan SM background in order to properly account for interference between signal and background. Published background estimates and observed events are then used to construct likelihood functions. It is straightforward to include correlations and/or systematic background uncer-

---

[7]The code can, however, take parton-level LHE events as input apart from SLHA files.

tainties when available.

DarkCast [93] is a public code for recasting dark photon search results; current bounds and future projections can be reinterpreted for arbitrary models with a new massive gauge field defined by its vector couplings to the SM fermions. Moreover, DarkCast provides a data-driven method for determining the hadronic decay rates for these new vector bosons at the GeV scale. The code is based on rescaling ratios of the production and decay rates, while also accounting for detector efficiencies due to the lifetime of the vector boson. It includes a comprehensive set of data from ATLAS, CMS and LHCb dark photon searches and a number of low energy (e.g. BaBar, Belle II) and beam-dump (e.g. NA62 and NA64) experiments. The LHC searches include dark photon resonance searches via data scouting methods [110–112] with future projections [113, 114], and Higgs-produced dark photon decays into prompt [115–117] and displaced [118] lepton-jet final states, as well as lepton-jet projections for the HL-LHC [119, 120].

Along similar lines, the DarkEFT package can be used to place limits on light dark sectors interacting with the SM through a heavier off-shell mediator in an effective "fermion portal" approach [104]. In addition to limits from LEP and most relevant intensity frontier experiments, the code includes the ATLAS 36 fb$^{-1}$ mono-jets limits [121] following the approach from [122]. Moreover, it allows one to evaluate prospects for FASER [123] and the potential reach of MATHUSLA [124].

## 3.2 Interpretation of measurements

Measurements of the type discussed in Section 2.2 can be and have been used to constrain BSM physics implemented both in EFT Lagrangians and in explicit simplified (or in some cases complete) BSM scenarios, and to reinterpret results in light of new SM predictions, without the need for approximate detector simulations. In the EFT case, the new physics appears in the form of anomalous effective couplings, which typically add or interfere with the SM at amplitude level, modifying measured distributions. This assumes that the new degrees of freedom of the model have masses above the maximum energy accessible at the LHC. In the case of simplified or complete scenarios, the new degrees of freedom may be accessible at LHC energies, and resonant bumps or other striking final state features may appear. However, in this case, interference effects are typically not considered although they may be important in some cases [92].

Measurements with a primarily SM focus such as the measurement of di- or tri-boson production rates [125–127] provide interpretations in terms of EFTs together with sufficient information to enable reinterpretation [128–130] outside of the experiments themselves. Simplified template cross-sections have proven useful for global SM EFT analyses [131] but care must be taken to assess potential biases caused by the use of a methodology reliant on Standard Model template fits to data to quantify high-energy modifications of observables from anomalous EFT couplings. Recent studies on Higgs boson data [132] have attempted to mitigate some of these model dependencies.[8] Published measurements [133,134] of Higgs boson production rates that are more model-independent have been combined and reinterpreted by the wider community without the need for specialised software to constrain CP-violating effects in Higgs couplings [135]. Such results illustrate the additional insights that are possible to achieve through interpretations of measurements, and what is not possible due to the lack of currently available public information. Beyond EFTs, studies [136] have demonstrated how differential cross-section measurements can be used to search for dijet resonances with competitive performance to dedicated resonance searches.

---

[8]See Section 2.2.7 for suggestions for further improvements in the presentation of the STXS results, which would allow for phenomenological analyses to study the remaining dependence on changes in the kinematic acceptance of the signal.

While the above measurements have a direct SM motivation, a recent proof-of-principle publication [137] has demonstrated an alternative approach where measurements are designed with BSM interpretations in mind. Observables sensitive to the presence of DM (or other invisible phenomena) are measured and data and auxiliary material and code are made public through HEPData and Rivet to enable reinterpretation of data by the wider community. Competitive constraints on various EFT, simplified, and complete DM models are documented in the publication using only publicly-available data/code to illustrate the power of interpretation of measurements compared to dedicated searches. These measurements are designed with interpretation in mind, by explicitly testing the model-independence of the methodology used to derive the results, and any limitations on use are discussed in the publication. Crucially, statistical and systematic correlation information between kinematic regions and different observables are published alongside the primary data, that can be exploited to enhance sensitivity over standard searches.

Measurements for which Rivet routines are provided can also be used by the Contur [61] package to derive exclusions for BSM models and parameters that would have led to a significant population of events within the fiducial phase space of the measurement. As the measurements currently included in Contur have all been shown to be consistent with the SM, by default Contur makes the assumption that the data are equal to the SM prediction, and uses the data uncertainties to evaluate the significance of any hypothetical deviation caused by the BSM model under consideration. Statistical correlations are avoided by dividing the measurements into orthogonal datasets (by running period and final state) and only taking the most sensitive bin within each such dataset. This also has the effect of avoiding the overestimation of the significance that would come from treating bins with highly correlated systematics as independent deviations. Currently Herwig [138] is used to generate all final states implied by a BSM model, with the model passed to Herwig via a UFO [139] directory. Several studies using Contur have been published, including one instigated during the recent 5th workshop of the LHC Re-Interpretation Forum [140]. These studies generally demonstrate the utility of the approach for rapid checking of the viability of a new model; see also e.g. Ref. [141]. They allow consideration of multiple final states simultaneously, which facilitates interpolation between different benchmark points with rather different phenomenology (see for example [142]). Where the datasets used by searches and the measurements in Contur are the same, the sensitivity is generally very similar; see e.g. the DM limits discussed in Ref. [143]. More recent developments of Contur include the correct use of correlated uncertainties when available in HEPData, the ability to use search routines from Rivet as well as measurements, the ability to use SM theory predictions as the background, instead of assuming that the data are identically equal to the SM, and tools for identification of the most sensitive contributing analyses through the parameter space [144].

On the flavour side, two different approaches are generally used. The first approach is model-independent, in which a set of Wilson coefficients is assumed and the coefficients are considered as independent parameters and fitted to the data. The second approach is model-specific, in which the observables are directly computed within a BSM scenario and constraints on the BSM parameters are obtained using the data. In the second approach it is possible to combine the results also with constraints from other sectors (collider and/or dark matter). In both approaches, correlations between experimental measurements are important, together with the theoretical correlations. Likelihoods for recasting based on flavour measurements in the model-independent approach are available in the FlavBit [145], flavio [146] and HEP-Fit [147] packages. FlavBit is the only package for both computing Wilson coefficients and carrying out a global fit. It is designed in the context of the GAMBIT framework, but can also be used in standalone form. The theoretical calculations of the Wilson coefficients and the relevant flavour physics observables in various models are done with SuperIso [148–150], and

the combined likelihoods for arbitrary combinations of the observables can then be obtained; in the latest version [151] this is via an interface to HEPLike [26].

# 4 Global fits to LHC data

Global fits are employed for various reasons. Sometimes, for example, a BSM model would not provide a statistically unambiguous experimental signal in a single channel, but only become evident by considering the totality of evidence in a coherent fashion. Also where there *is* an obvious signal in a single channel—and even more importantly in the case of actual evidence for a new signal— one would immediately wish to know how limits from other searches and measurements break the degeneracy of the family of models compatible with the positive result. It is clear that such combinations of experimental results provide a more complete picture than individual ones in isolation.

The tension (or agreement) with fruitless searches is then also taken into account. A secondary goal may be to identify targets for future searches of "most likely parameter regions", where the most sensitive signals can be identified. A number of global fits to LHC data have been performed in the context of BSM models in recent years, typically using public reinterpretation packages. Global fits are performed in the context of a full model rather than that of a simplified model, since it is in full models where correlations between different data (including search and indirect data such as electroweak fits) are present and relevant. The problem of translating global fits in one full model into fits in another full model is dealt with by public global fitting packages such as GAMBIT [99,100], MasterCode [152–154] and HEPFit [147].

Both high-scale [155] and low-scale parameterisations [12,156] of the MSSM have been studied using the ColliderBit machinery within GAMBIT. The most recent of these [12] reinterprets 12 different ATLAS and CMS simplified model searches for electroweak production of sparticles based on $36\,\mathrm{fb}^{-1}$ of $13\,\mathrm{TeV}$ data, in terms of (essentially) an MSSM electroweakino EFT, finding a $> 3\sigma$ (local) preference for light charginos and neutralinos.

Similar avatars of the MSSM have been investigated by the MasterCode collaboration [152, 153], along with DM simplified models [154]. These studies reinterpret a variety of ATLAS and CMS searches for supersymmetric particles and DM, heavy Higgs bosons, and new dijet resonances. Likelihood functions for SUSY searches are modelled as in the fastlim code, which takes a similar approach as SModelS.

Effective field theories as previously described can be an efficient way to describe physics and parameterise global fits to data, despite the fact that they may have limitations in their applicability. In the EFT picture, the low-energy effects of heavier new physics are captured from a bottom-up perspective, through a systematic organisation of departures from the SM as an expansion in the inverse of the new physics scale. These effects are described in terms of higher-dimensional operators, whose running to the high scale and matching to any UV-complete theory allows for the reinterpretation of any possible departure from the SM in the framework of several explicit BSM models at once.

Given the SM field content and gauge interactions, the leading effects are assumed to arise from dimension-six operators in the SM fields that are supplemented to the SM Lagrangian [157–159], which defines the so-called SMEFT framework. These parametrise fits to current electroweak and Higgs data, which facilitate the extraction of Higgs couplings (see e.g. the Run 2 analyses of ref. [78,131,160–163]), and provides a framework to parameterise the searches for new physics that is testable from the expected correlations between the expected signatures. The SMEFT framework can be used to interpret data beyond Higgs and electroweak physics. For example, possible new phenomena in the top quark sector can be included in the SMEFT scenario, see e.g. Refs. [162,164]. Moreover, after being evolved to

the scale of the $b$-quark mass, the same SMEFT framework allows for the analysis of the recent data based on $B$-meson decays that are in tension with the SM predictions. In particular, combined data on rare decays involving the SM particles $\bar{b}s\bar{\mu}\mu + H.c.$ and decays involving $\bar{b}c\bar{\tau}\nu + H.c.$ are currently at (or are at more than) a $3\sigma$ level of tension. Contributions from new physics to flavour-violating operators involving these external particles thus appear to be favoured by global fits to the flavour data. These have been parameterised by particular 4-fermion dimension-6 operators (see e.g. Refs. [165–169] for a few recent results). Simplified models, see e.g. Ref. [170], may then be invoked to postdict the particular values of the BSM EFT operators implied by the global fits. (A vast literature exists using both simplified and non-simplified models.) Once these models have been matched to the EFT operators implied by data, one has a parameterisation of the regions of model parameter space that provide a good global fit. These regions can then be examined for direct search constraints (in the examples mentioned above, $Z'$ bosons with family dependent couplings were invoked to explain the $\bar{b}s\bar{\mu}\mu$ data; LHC searches for $pp \to Z' \to \mu^+\mu^-$ [10] then constrain the models). As usual, the prompt publication of direct search data in HEPData facilitated its accurate re-use.

## 5   Summary

Since the time of the previous recommendation document [1], the increase in information provided for both searches and measurements has significantly improved the accuracy with which LHC experimental results can be reinterpreted. The wide availability of this information has energised interactions between experiment and theory, leading to the creation of a number of dedicated and sophisticated reinterpretation frameworks. These have led in turn to a proliferation of phenomenological studies based on the reinterpretation of specific searches and measurements, as well as detailed global fits to reinterpreted data from multiple channels.

The emphasis has now moved from advocating the provision of reinterpretable data by experiments *a priori*, to ensuring their ubiquity, comprehensiveness, and presentational uniformity, in order to take re-use to the next level. From the experimental collaboration and individual experimenter point of view, re-use means a longer legacy for analyses, as well as compliance with ever stricter requirements of data-publication and reusability for publicly funded research. In Section 2 of this document, we discussed a list of specific recommendations for particular data types.

Provision of auxiliary data is often patchy even when the experiment has a standard. We therefore suggest more generally that the experiments consider more formal enforcement as a pre-publication step; this information is as scientifically important as the more recognised document published in a journal. If made mandatory, experience shows that effort will be found (as it is for many more challenging tasks during analysis and publication approval). To ensure the preservation at long term, beyond the lifetime of the experiment, we recommend that all material be provided on a site dedicated to data preservation, i.e. HEPData or Zenodo, in addition to experiments' own analysis-resource web pages. These public platforms are closely connected to the efforts of Inspire, which has pioneered the collection and preservation of metadata. Their work made it possible to publish data files, analysis codes and other auxiliary material and data in a searchable, properly versioned and separately citable form—we are grateful to and fully supportive of these important efforts.

Our conclusions are summarised as follows:

1. It is crucial that numerical analysis data to enable re-use, both in searches and in measurements, be made promptly available in digitised electronic form, preferably via the established and stable HEPData and Zenodo systems. As analysis person-power often

disappears rapidly after publication, we strongly recommend that experimental collaborations make provision of data to enable re-use a mandatory step before journal publication. To minimise disruption of the scientific conversation, such a requirement cannot be imposed strictly on conference notes, but we argue that the "damage" of making preliminary data available (with a clear "health warning") falls primarily on the users, and that a potentially greater risk is associated to figure digitisation in the absence of official numbers.

2. In Run 2 and beyond it is essential, both for reasons of precision and of stability of global fits, that correlation information also be provided in a readily re-usable format: pros and cons of possible compact representations are discussed in Section 2 and Appendix A of this document. Moreover, going beyond the provision of simplified correlation information, we strongly encourage the publication of full likelihoods.

3. Additional digitised information is also needed for accurate reproduction, validation, and extension of phenomenological studies in experimental analysis papers: in particular, exact BSM model files, MC generator steering files, cut-flow tables, SM-background estimates, and numerical limit/efficiency parameter map data. We encourage publication of MC datasets, at least at parton level, but ideally identical to those used in the experiment, to aid validation. Any modifications made to original BSM models, such as event clipping or other unitarisation procedures, should be documented.

4. We note with gratitude the initiatives taken toward publication of full likelihood, Open Data, Rivet routine publication, and other comprehensive analysis preservation strategies. Building on this, we encourage full coverage of relevant experimental measurement analyses by promptly published Rivet analysis routines, full publication of search analysis routines and framework code, and public sharing of MC samples where possible. Formal incorporation of these initiatives into publication processes may be essential to attain the necessary level of coverage.

5. More complete publication of full-detail experimental data, in the forms of Open Data and forensic analysis code preservation via container images, is also very welcome. We encourage universal uptake of these approaches across the LHC experiments, with agreement on common embargo periods before public release – noting the precedents for early full-release of data established by the astrophysics community and particularly LIGO [171]. The nature of these "heavy" formats for data release means that a mixed strategy of more lightweight approaches based on fast-simulation and similar are an essential complement to enable effective re-use of analysis results beyond the experimental collaborations.

6. In measurements, subtraction of irreducible backgrounds can be problematic for interpretations beyond the SM. Moreover, the presentation of cross-sections (or limits thereon) relative to the cross-section predictions of a model, a.k.a. signal strengths, complicates reinterpretations in different models. Complications also arise when results have to be re-evaluated because more precise theoretical predictions become available. We, therefore, recommend that *in addition* to process-specific subtractions and model-specific interpretations (including SM, cf. STXS), cross-sections be published at a purely fiducial final-state level: that is, best estimates of true observable event rates obtained from experimental observations.

7. New techniques such as use of unbinned fits and machine-learning algorithms are entering BSM searches and to a lesser extent measurements. These pose new challenges for reinterpretation and re-use of analysis data, and community-wide dialogue is needed to

ensure that such approaches remain consistent with goals of long-term analysis re-use and impact.

8. For many models interference effects between signals of BSM physics and SM background can be important, in particular in the context of searches for broad resonances. It will be crucial for the reinterpretation of experimental results to develop new methods to treat these model-dependent effects in a general way.

9. Theorists should let the experimental collaborations know about the use of analyses in reinterpretations, as this can be important feedback. We also stress here the importance of labelling correctly the use of preliminary results, although it is always recommended to use published results.

10. Last but not least, we strongly encourage theorists to follow the same reproducibility requirements as we ask them from the experiments. Concretely, we recommend that codes developed for reinterpretation be made public, e.g. on GitHub or by integration in an existing public framework, and that analysis inputs and results be made available in digital form; a dedicated Zenodo community[9] exists for this purpose.

## A  Technical recommendations on how to publish correlations

Reporting of correlations is established via HEPData, either as auxiliary data files or (more helpfully) as directly encoded datasets. As the numerical correlation format choice is currently a free choice for each analysis, this is an area where more standardisation would greatly assist re-use. We encourage use of the "orthogonal error sources decomposition" format,[10] directly attached to the signal-region bins in yield tables, rather than separate matrix tables. The motivations for this recommendation are:

– they are more easily (programmatically) identified with primary data, as they are part of the same dataset;

– matrix tables need a mechanism to indicate whether they represent absolute covariance or relative correlation values; and

– error sources enable the encoding of error asymmetries, which are lost in the implicitly symmetric Gaussian construction of a covariance.

We further recommend providing correlation information in the form of orthogonal error sources and developing the ability of HEPData and downstream tools to render this data into correlation or covariance figures and tables.

Should a matrix format be necessary, we recommend use of new standard identifiers in the correlation dataset's header to identify the primary dataset to which it relates (e.g. primary:tab2), and to clarify if the numbers are dimensionful covariances (e.g. cov) or unit-normalised correlations (e.g. corr). As correlations can be computed from covariances alone, but the reverse is not true, we suggest covariance data as the preferred form if only one is provided.

Currently, correlation matrix data is often provided as auxiliary files in the ROOT format, with great variation in the internal path structure and the type of the "leaf" data object (sometimes the graphical object used to make a 2D "heatmap" figure is stored, rather than

---

[9]https://zenodo.org/communities/lhc-recasting/

[10]Such a decomposition is always possible, including from both effective error-correlation formalisms like Simplified Likelihood, and from elementary nuisances (which will typically only be orthogonal for pre-fit numbers).

the raw data required for likelihood construction). Native HEPData data objects are easier to work with, but should ROOT files be used we recommend that they be located in the "root directory" of the tree, be named systematically as either cov or corr (cf. discussion above), and are uniformly the numerical TH2s rather than graphical objects. We note also the rise in popularity of Python data-science tools like NUMPY and PANDAS within HEP; the same considerations largely apply to data published as auxiliary files in e.g. the numpy text format, or the HDF5 [172] format.

Whatever format it will be provided in, it is essential that enough precision be used for the reporting that the associated covariance matrix $\Sigma_{ij}$ is invertible. This has often not been the case with previous reporting via HEPData, creating issues when attempting to e.g. evaluate the correlated $\chi^2 = \sum_{ij} \Delta x_i \Sigma_{ij}^{-1} \Delta x_j$ statistic. Where distributions have been unit-normalised, a covariance matrix over all bins is not invertible, and if extended over multiple independently unit-normalised distributions the procedure to achieve a correct inversion is non-obvious and may require *pre*-normalisation application of the correlations. In this situation, the analysis should provide explicit instructions for how to use the provided correlation data to construct a representative goodness of fit measure.

Finally we note the existence of the "next-to-simplified likelihood" [13], a simple method to encode asymmetry information into correlations via publication of only $N_{\text{bins}}$ additional numbers. This, or public full-likelihood encodings in a standard format, are particularly important for low-statistics bins at the kinematic limits of the collider: they will hence become increasingly essential as the high-luminosity LHC runs progress.

# Acknowledgements

We thank the LHC Physics Centers at CERN and Fermilab, the CERN and Fermilab Theory Departments, as well as the IoP London and the IPPP Durham, UK, for support in the organisation of workshops associated to this work. We also acknowledge the inspiring atmosphere at the Les Houches "Physics at TeV Colliders" workshops, where the reinterpretation of LHC results remains an active topic.

The editors of this report were supported in part by the European Union as part of the Marie Sklodowska-Curie Innovative Training Network MCnetITN3 (grant agreement no. 722104), by the IN2P3/CNRS under the project "LHC-iTools", by STFC grants ST/P000681/1, ST/K00414X/1, ST/N000838/1, ST/P000762/1, ST/N003985/1, ST/M005437/1, and by FAPESP under the project grant 2015/20570-1. We acknowledge moreover partial support by The Royal Society, University Research Fellowship grant UF160548 (AB); the F.R.S.-FNRS (JH); the DFG Emmy Noether Grant No. KA 4662/1-1 (FK). the Austrian Science Fund, Elise-Richter grant V592-N27 (SuK); and the National Research Foundation of Korea, NRF, contract NRF-2008-00460 (SS).

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
