# Peer review of "Reinterpretation of LHC Results for New Physics: Status and Recommendations after Run 2"

_SciPost Physics, doi:SciPost Phys. 9, 022 (2020)_

## Round 2 · Referee Report · Anonymous (Referee 1) · 2020-5-3

Strengths

They are described clearly in the report.

Weaknesses

They are described clearly in the report.

Report

I find this document a truly excellent and very useful piece of work. It can (and probably will) serve as a key reference for the Experimental Collaborations on making public all the necessary information needed for the proper re-interpretations of LHC results. As such I have minor comments mainly targeted to making communication of information more clear and explicit.

General Comment : Perhaps what is lacking in general in the paper is a detailed description of a key selected example (of the many cited) from each category (searches and measurements) where facilitated reinterpretations really made a significant impact in the field. I understand this is difficult, but if it can be formed and presented it would be very useful.

I.

Page 7 second paragraph : Perhaps more emphasis should be given on the scientific – physics reasons why reinterpretations are very important, which should therefore be listed first.

II.

A. Searches

Section 1
Page 10 last paragraph, Page 11 first paragraph, and Page 11 second paragraph at the end : it would be perhaps good to summarize in a clear manner (with bullets for example) what the explicit recommendations are for this category. This is also true for the rest of the sections in the document (see some examples below). A good example is Section 5 where the recommendations are clearly summarized at the end of the section.

Section 2
Page 11 first paragraph, page 12 second paragraph : here the recommendations are summarized in the Appendix, maybe the same should be done for all sections (see previous comment), or clearly in the text. In any case a clear and homogenized approach for the entire document should be followed.

Section 3
Again it is hard to find in the text all recommendation, so it would be very helpful to follow an approach outlined above for section 1 and organize clearly all recommendations.

Section 5 :
Despite the fact that this section clearly summarizes most recommendations on page 16, there are still recommendation after the list, for example on the second paragraph of page 16 in the middle, which would be good to be added to the list as well.

Page 19 last paragraph: I think the comment about the MC samples not being investments of intellectual effort, is neither correct [at least not in all cases], nor necessary to make the point. Generation of MC samples does often include a lot of effort in order to use the appropriate tunes determined by data fits, to communicate issues with the authors of the MC generators and work together to fix them, and certainly also involves the reconstruction, calibration, identification, selections of physics objects of each experiment, and many times the weighting with the so called "scale factors" coming from differences between experimental data and simulation in terms of trigger efficiencies, b-tagging efficiencies etc. Hence, making public simulation samples is sometimes a delicate and not as straightforward as it might seem procedure that needs to many details to be properly documented and made public as well. As such the MC distributions at reconstruction level often contain a lot of intellectual effort, and are incorporating specifics of each experiment. Having said that, I agree they should be shared when possible.

Section 8

Pseudo-code cannot easily be provided in the case of ever-growing ML method usage in searches, that need the specific distributions of many (sometimes hundreds) of analysis low and high level quantities to be computed. This is discussed in section 10 but perhaps some brief discussion in this section, and a pointer to the discussion in section 10, should be made given number of analyses that use this techniques now, and the fact they will be growing in the near future. This point is also nicely mentioned in the summary (point 7) and is it a very important one, hence should be emphasized more in the main body of the text as well when and where possible.

B. Measurements

-General comment : Again, excellent suggestions are sometimes lost in the text, so a clear summary in appendices and/or a bulleted line approach would be very helpful to be followed.

No specific suggestions or comments for sections III and IV, very nicely written and with a lot of detail.

Requested changes

They are described in the report.

  • validity: top
  • significance: high
  • originality: high
  • clarity: high
  • formatting: excellent
  • grammar: excellent

Author:  Andy Buckley  on 2020-07-22  [id 898]

(in reply to Report 1 on 2020-05-03)

Our thanks for the very constructive inputs, and apologies for the time delay in responding: a combination of Covid-19 effects, and a large authorship with which to consult. We have implemented the majority of suggestions, and just respond here on a few non-trivial points:

General Comment : Perhaps what is lacking in general in the paper is a detailed description of a key selected example (of the many cited) from each category (searches and measurements) where facilitated reinterpretations really made a significant impact in the field. I understand this is difficult, but if it can be formed and presented it would be very useful.

We understand the motivation and interest of this request, but highlighting one example would create a bias and do injustice to the many reinterpretation efforts that exist and give feedback to the experiments on different fronts. We hence prefer not to do this, despite the clear best intentions of the request.

Page 7 second paragraph : Perhaps more emphasis should be given on the scientific – physics reasons why reinterpretations are very important, which should therefore be listed first.

In order to focus more strongly on the benefit to physics of reinterpretation studies, we have replaced the first sentence in this paragraph with the following text: “In order to determine the implications of LHC data for a broad range of theories, experimental analyses must be reinterpretable in terms of theories not considered in the original analysis publication. This reinterpretation process, also known as ``recasting'', makes it possible for the community as a whole to test a much larger variety of theories using the LHC than would be possible purely within the experimental collaborations. This also makes it possible for phenomenologists to give detailed feedback on the original analyses, and to better suggest promising new avenues of experimental analysis.” We have also re-ordered the benefits to experiments listed in the following (third) paragraph, so that the first benefit listed is the most physically-relevant one (pointing experiments to promising new BSM models / parameter combinations).

Page 10 last paragraph, Page 11 first paragraph, and Page 11 second paragraph at the end : it would be perhaps good to summarize in a clear manner (with bullets for example) what the explicit recommendations are for this category. This is also true for the rest of the sections in the document (see some examples below). A good example is Section 5 where the recommendations are clearly summarized at the end of the section.

We have added the following bulleted summary to Sec II.A.1: In summary, we recommend that all experimental searches provide: 1. estimates of background rates in all signal regions, broken down into as many separate process contributions as possible, and 2. where backgrounds are fitted, full details of the fits: functional forms, fit ranges, fit procedures, best fits and uncertainties. We have not added such summaries everywhere in the document, as often there are complex considerations to be taken into account in order to correctly follow the recommendations, so relying on such summaries would be rather misleading.

Page 11 first paragraph, page 12 second paragraph : here the recommendations are summarized in the Appendix, maybe the same should be done for all sections (see previous comment), or clearly in the text. In any case a clear and homogenized approach for the entire document should be followed.

The discussion of how to publish correlations (on HEPData) is done apart (in an appendix) because it goes into much more technical detail than the other subsections. While integrating this appendix into the main body of the text would be possible, on reflection we preferred to retain it in appendix form for the flow of text, and rather than put all recommendations into appendices they have been clarified in bullet form at the end of each section.

Again it is hard to find in the text all recommendations, so it would be very helpful to follow an approach outlined above for section 1 and organize clearly all recommendations.

We have added the following bulleted summary to Sec II.A.3: We recommend that each experimental analysis identifies a corresponding set of recommended efficiencies and resolution functions for its reinterpretation, and provides them in the following forms if they are not already publicly available: Resolutions: Delphes cards, functional parameterisations with relevant parameters (for simple functions), or code snippets (for more complex functions). Efficiencies: tabulated eciencies in the kinematic variables most relevant for the analysis in question, in HEPData format (not ROOT format). Preferably broken down into as many sub-efficiencies (trigger, ID, etc) as possible, and given at object rather than event level.

Despite the fact that this section clearly summarizes most recommendations on page 16, there are still recommendation after the list, for example on the second paragraph of page 16 in the middle, which would be good to be added to the list as well. Done: We have extended item 3 and added a new item 6 to clearly summarize all recommendations of this subsection. We also rephrased the remark in the middle of the paragraph after the numbered items to make it clear that this is a comment and not a recommendation.

Page 19 last paragraph: I think the comment about the MC samples not being investments of intellectual effort, is neither correct [at least not in all cases], nor necessary to make the point. Generation of MC samples does often include a lot of effort in order to use the appropriate tunes determined by data fits, to communicate issues with the authors of the MC generators and work together to fix them, and certainly also involves the reconstruction, calibration, identification, selections of physics objects of each experiment, and many times the weighting with the so called "scale factors" coming from differences between experimental data and simulation in terms of trigger efficiencies, b-tagging efficiencies etc. Hence, making public simulation samples is sometimes a delicate and not as straightforward as it might seem procedure that needs to many details to be properly documented and made public as well. As such the MC distributions at reconstruction level often contain a lot of intellectual effort, and are incorporating specifics of each experiment. Having said that, I agree they should be shared when possible.

There seems to be a miscommunication here: we are talking purely about truth-level MC samples (and hence reconstruction, reweighting, efficiencies, etc. do not apply. It would be useful for the reinterpretation community to also have access to reco-level samples, analysis workflows that apply reweighting, etc., but these definitely are experiment IP, and fall under the Recast and Open Data initiatives. Tunes and fixes to MC generators are propagated back to the community, so are already effectively public (and this is a good thing). We do not mean to say that designing correct and efficient MC process configurations is devoid of IP — if it were, there would be little point in making it public — but that the vast majority of it is not “commercially confidential” to the experiments. We will rephrase to remove the trigger phrase.

Pseudo-code cannot easily be provided in the case of ever-growing ML method usage in searches, that need the specific distributions of many (sometimes hundreds) of analysis low and high level quantities to be computed. This is discussed in section 10 but perhaps some brief discussion in this section, and a pointer to the discussion in section 10, should be made given number of analyses that use this techniques now, and the fact they will be growing in the near future. This point is also nicely mentioned in the summary (point 7) and is it a very important one, hence should be emphasized more in the main body of the text as well when and where possible.

We entirely agree and thank the referee for raising this point. We have added a short paragraph at the end of section II.A.8 following the referee’s suggestion.

---

## Round 2 · Referee Report · Anonymous (Referee 2) · 2020-5-4

Strengths

1) Well-structured. 2) Timely. 3) Essential to the field. 4) Well-written and clear. 5) Comprehensive in detail and scope. 6) Definitive.

Weaknesses

1) It's not so much a weakness of the paper but of the format, but there are a few places that would benefit from a paragraph and/or figure illustrating a concrete example of one of the methods, techniques, or tools described, beyond a citation. The intent of this paper being what it is, however, makes this not really appropriate, so it is mentioned here not so much as a weakness than as a compliment: It makes we want to know more about the techniques I'm not so familiar with and follow the citations.

Report

This is a comprehensive, well-structured, clearly-written paper, essential to the field and timely. It leaves very little to be desired because everything is included. The editors and authors are to be commended, and I look forward to seeing the bolder recommendations adopted and executed by the experiments, such as uniformly providing numerical data for preliminary results rather than only "published" results. Congratulations on an excellent, definitive document.

Requested changes

None.

  • validity: -
  • significance: -
  • originality: -
  • clarity: -
  • formatting: -
  • grammar: -

Author:  Andy Buckley  on 2020-07-22  [id 899]

(in reply to Report 2 on 2020-05-04)
Category:
remark

Our thanks to the reviewer for their complimentary words: we share the same aspirations.

---

## Round 2 · Referee Report · Anonymous (Referee 3) · 2020-5-24

Strengths

  • clearly defined purpose
  • coordinated effort of a significant fraction of te experts in the community
  • very useful practical guidelines to ensure that the data collected by current collider experiments can be used by others, and can be stored for the future
  • well-written and well-structured

Weaknesses

  • ideally the priorities of the community among the many recommendations could be outlined a bit more clearly

Report

As requested by the editor, I reviewed the document from the viewpoint of a general BSM theorist. I am by no means an expert on the technical details of state-of-the-art re-interpretation approaches, so in my assessment I take the viewpoint of a critical "external observer”.

Leaving aside the introduction, summary and the technical appendix, the document consists of three main parts, the sections II-IV. These are devoted to the information provided by the experiment (II), a comparison of interpretation methods (III) and a brief discussion of global fits (IV).

Section II makes up most of the document. It mainly consists of detailed discussions of the data to be provided in BSM searches (A) and SM measurements (B), followed by a shorter discussion of open data strategies (C). I find the presentation very concise and clear. Different types of information are discussed one by one. This includes primary data and background estimates as well as derived quantities (such as likelihoods and correlations), and details of the analysis (such statistical methods, efficiencies, smearing functions used, and underlying theory assumptions). In each case, useful recommendations for the community are made on how to publish this information. Clear motivations are given for each recommendation, and positive examples in the literature for good practice are pointed out. While some recommendations are rather generic and simply reflect "common sense", others concern the specific format in which information should be published. As a theorist I also appreciate the comments on assumptions, e.g. in simplified model analyses. Overall, I believe that this section provides many useful guidelines and specific suggestions for the experimental community (and to some degree also theorists).

In section III a list of various different reinterpretation methods is given. I do not have much experience with these, and I am not in a position to comment on the completeness or correctness of the information presented here. But I do think that this section is a very helpful summary and list of references for anyone who wants to start working on re-interpretations.

I find the discussion of global fits a bit weaker than the other two sections. In large parts it comprises a list of what has been done, which is of course useful. But it is lacking specific recommendations how to deal with the main issues of global fits, such as: the strong dependence of the “most likely parameter region” on the parameterization and choice of priors, or the difficulty to translate result obtained in specific full models to other models. In any case, the section represents a useful list of references.

In section V the authors summarize the most important recommendations regarding the type of information that should be published and the form in which this should be done. They also encourage the experimental collaborations to make data available before journal publication and call for close interactions between the experimental and theoretical communities.

My overall impression is that the document is well-organized and well-written. The amount of genuinely original material is very limited, but this is normal for an article of this kind. I believe that the document provides a helpful guideline for the community and should be published. If all these suggestions are adapted by the collaborations, this would certainly be extremely useful. My only somewhat critical comment is that it may be helpful to indicate the priorities in the long list of recommendations a bit more clearly. However, I of course understand that this may be difficult in a document with so many authors who may have different opinions about these priorities.

Requested changes

see above

  • validity: high
  • significance: high
  • originality: high
  • clarity: high
  • formatting: excellent
  • grammar: excellent

Author:  Andy Buckley  on 2020-07-22  [id 900]

(in reply to Report 3 on 2020-05-24)
Category:
remark

Our thanks to the referee for very helpful comments, which echoed another call for clearer presentation of the concrete recommendations, which are addressed in the new version (v3). We are glad that they think the document a useful guide for both non-specialist BSM theorists and experiment(alist)s. Our responses to specific critiques follow:

I find the discussion of global fits a bit weaker than the other two sections. In large parts it comprises a list of what has been done, which is of course useful. But it is lacking specific recommendations how to deal with the main issues of global fits, such as: the strong dependence of the “most likely parameter region” on the parameterization and choice of priors, or the difficulty to translate result obtained in specific full models to other models. In any case, the section represents a useful list of references.

The intended function of this section was not actually to give recommendations, but rather to give examples of comprehensive analyses that have been enabled by the data provided by the experiments for reinterpretation. We have made this clearer in the ‘paper roadmap’ paragraph at the end of Sec I. As to the two “main issues” with global fits that the referee identifies, the second (translating global fit results obtained in specific full models to other models) is essentially the issue addressed by global fitting packages such as GAMBIT and HEPfit. We have added the following sentence to the end of the second paragraph of Sec IV in order to point this out: “The problem of translating global fits in one full model into fits in another full model is dealt with by public global fitting packages such as GAMBIT [x,x], MasterCode [x,x] and HEPFit [x]”. It is not generally accepted that the first issue actually constitutes a problem. The “most likely parameter region” is only well defined in a Bayesian sense, in which case, prior/parameterisation dependence of results is a completely natural and normal feature of the analysis method, and provides information as to the degree of fine-tuning in different parts of the parameter space, as well as the degree to which the data are at all informative. Complementary and prior/parameterisation-independent information can be gained from a purely profile-likelihood analysis, which shows to what extent different parameter combinations are compatible with the existing data. If one is bothered by or otherwise unsure of how to correctly interpret Bayesian results, one can instead focus on the profile likelihoods, and the information that they provide.

My only somewhat critical comment is that it may be helpful to indicate the priorities in the long list of recommendations a bit more clearly. However, I of course understand that this may be difficult in a document with so many authors who may have different opinions about these priorities.

The priorities the referee asks for depend a lot on the (kind of) analysis, and it would go too far to discuss these aspects in this report - even more so as for reliable reproducibility, the aim is that ALL recommendations be followed. Moreover, it is beyond the mandate of the Reinterpretation Forum to prioritize, e.g., between REANA/RECAST and Open Data efforts. These aspects have to be decided by the experimental collaborations.
One clear priority, however, is to have all material for recasting in numerical form, preferably on HEPData, and we think that this is made sufficiently clear in the conclusions. For clarity, the new version of the manuscript re-presents the key recommendations via lists at the end of each section.

---

## Round 3 · Author Response

This version incorporates the feedback from the referees, which was broadly positive but encouraged clearer presentation of the concrete recommendations, and finessing of some text. Thanks for the constructive process and your patience.

---

## Round 3 · List of Changes

Introduction of end-of-section lists of concrete recommendations; refinements to discussions of motivation, pseudocode, and MC sample sharing; small updates to MadAnalysis and Contur text and citations.

---

## Editorial Decision

published